# Tumor-induced MDSC act via remote control to inhibit L-selectin-dependent adaptive immunity in lymph nodes

Amy W Ku[1†], Jason B Muhitch[1,2†], Colin A Powers[3], Michael Diehl[1], Minhyung Kim[3], Daniel T Fisher[1,3], Anand P Sharda[2], Virginia K Clements[4], Kieran O'Loughlin[5], Hans Minderman[5], Michelle N Messmer[1], Jing Ma[6], Joseph J Skitzki[3], Douglas A Steeber[7], Bruce Walcheck[6], Suzanne Ostrand-Rosenberg[4], Scott I Abrams[1], Sharon S Evans[1*]

[1]Department of Immunology, Roswell Park Cancer Institute, Buffalo, United States; [2]Department of Urology, Roswell Park Cancer Institute, Buffalo, United States; [3]Department of Surgical Oncology, Roswell Park Cancer Institute, Buffalo, United States; [4]Department of Biological Sciences, University of Maryland Baltimore County, Baltimore, United States; [5]Flow and Image Cytometry, Roswell Park Cancer Institute, Buffalo, United States; [6]Department of Veterinary and Biomedical Sciences, University of Minnesota, St. Paul, United States; [7]Department of Biological Sciences, University of Wisconsin-Milwaukee, Milwaukee, United States

*For correspondence: sharon.evans@roswellpark.org

†These authors contributed equally to this work

Competing interests: The authors declare that no competing interests exist.

**Abstract** Myeloid-derived suppressor cells (MDSC) contribute to an immunosuppressive network that drives cancer escape by disabling T cell adaptive immunity. The prevailing view is that MDSC-mediated immunosuppression is restricted to tissues where MDSC co-mingle with T cells. Here we show that splenic or, unexpectedly, blood-borne MDSC execute far-reaching immune suppression by reducing expression of the L-selectin lymph node (LN) homing receptor on naïve T and B cells. MDSC-induced L-selectin loss occurs through a contact-dependent, post-transcriptional mechanism that is independent of the major L-selectin sheddase, ADAM17, but results in significant elevation of circulating L-selectin in tumor-bearing mice. Even moderate deficits in L-selectin expression disrupt T cell trafficking to distant LN. Furthermore, T cells preconditioned by MDSC have diminished responses to subsequent antigen exposure, which in conjunction with reduced trafficking, severely restricts antigen-driven expansion in widely-dispersed LN. These results establish novel mechanisms for MDSC-mediated immunosuppression that have unanticipated implications for systemic cancer immunity.

## Introduction

Myeloid-derived suppressor cells (MDSC) have emerged as important immune regulators in cross-disciplinary fields including cancer biology, immunotherapy, chronic infection, and autoimmunity (*Cripps and Gorham, 2011*; *Gabrilovich et al., 2012*; *Goh et al., 2013*; *Talmadge and Gabrilovich, 2013*; *Crook and Liu, 2014*). MDSC have been most extensively characterized in the context of cancer where they thwart antitumor adaptive immunity (*Gabrilovich et al., 2012*). MDSC accumulate throughout cancer progression and are linked to poor clinical outcomes (*Liu et al., 2010*; *Waight et al., 2013*) as well as resistance to chemotherapy, radiation, and immunotherapy in murine tumor systems (*Acharyya et al., 2012*; *Xu et al., 2013*; *Alizadeh et al., 2014*). MDSC exert hallmark immunosuppressive activities via production of arginase, reactive oxygen and nitrogen species, and

indolamine 2,3-dioxygenase that locally block activation of tumor-specific T cells (*Gabrilovich et al., 2012*). These immature myeloid cells further contribute to tumor immune evasion by expressing immunosuppressive molecules such as programmed death-ligand 1 (PD-L1) (*Youn et al., 2008*) and by supporting the function of immunosuppressive regulatory T cells (Treg) (*Huang et al., 2006*) and M2 macrophages (*Sinha et al., 2007*; *Beury et al., 2014*). Evidence that the MDSC-T cell suppressive axis is enacted by short-lived, contact-dependent mechanisms (*Sinha et al., 2007*; *Gabrilovich et al., 2012*; *Ostrand-Rosenberg et al., 2012*) supports the prevailing view that suppressive effector functions are mainly restricted to tissues where MDSC and T cells both localize.

The majority of studies have focused on MDSC-enriched tumors and splenic reservoirs as the major locale where MDSC execute suppression of local T cell function (*Gabrilovich et al., 2012*). MDSC are also abundant in the circulation of tumor-bearing mice and cancer patients (*Ostrand-Rosenberg and Sinha, 2009*) although it is not known if MDSC in the blood compartment mediate immunosuppression in situ. In contrast, MDSC are rare in lymph nodes (LN) (*Ostrand-Rosenberg and Sinha, 2009*; *Garcia et al., 2014*), and thus, their suppressive roles at these critical sites of immune priming are largely overlooked. Our prior work demonstrating that MDSC partially downregulate expression of the L-selectin LN homing receptor on naïve T cells (*Hanson et al., 2009*; *Parker et al., 2014*) suggested that MDSC might interfere with T cell function by preventing access to the LN microenvironment. L-selectin–mediated tethering and rolling within vessel walls is a prerequisite for trafficking of naïve T and B cells across gateway high endothelial venules (HEV) in LN (*Girard et al., 2012*; *Evans et al., 2015*). Efficient trafficking at HEV increases the probability that activating signals are delivered to specific-antigen restricted naïve T and B lymphocytes existing at a frequency of only ~1 in $10^5$–$10^6$ in mice and humans (*Oshiba et al., 1994*; *Jenkins et al., 2010*). In murine tumor models, MDSC are associated with partial reduction of L-selectin on naive T cells that can be restored upon MDSC depletion using gemcitabine-based chemotherapy (*Hanson et al., 2009*). However, the biological implications of L-selectin down-modulation cannot be inferred solely from expression analysis since the high L-selectin density normally present on leukocytes (~50,000–100,000 molecules per cell) (*Kishimoto et al., 1989*; *Simon et al., 1992*) could theoretically buffer against moderate fluctuations in expression during homing. In the present study we tested the hypothesis that MDSC are capable of systemic immunosuppression by investigating: (*a*) the spatiotemporally-regulated mechanisms underlying MDSC-driven L-selectin down-modulation in T cells, (*b*) whether L-selectin loss extends to B cells which are not validated MDSC targets in cancer, (*c*) if moderate L-selectin loss is sufficient to compromise lymphocyte trafficking and antigen-induced priming within the intranodal compartment.

Here we report that MDSC cause far-reaching immune suppression by downregulating L-selectin at discrete anatomical sites in murine tumor models. We determined that MDSC function through a contact-dependent mechanism independent of the major L-selectin sheddase, a disintegrin and metalloprotease (ADAM) 17, to target L-selectin loss exclusively on naïve CD4[+] and CD8[+] T cells located in close proximity within the splenic compartment. Surprisingly, this mechanism also takes place on both T and B cells as they circulate together with MDSC within the intravascular space. Our studies further show that even modest MDSC-driven L-selectin down-modulation is sufficient to profoundly reduce homing and antigen-dependent activation of naïve CD8[+] T cells in LN which is attributed to decreases in the quality of L-selectin-dependent rolling interactions within HEV. Additionally, we identify a role for MDSC preconditioning of T cells in the spleen or blood that reduces responsiveness to antigen outside these organ sites. These findings support a model in which MDSC act at remote tissue compartments to exert wide-spread, systemic immune suppression in distant LN, thus having broad implications to cancer immunity and inflammatory disease processes.

## Results

### Spatiotemporal correlation between L-selectin loss and MDSC co-localization with naïve T cells in the splenic compartment

To gain insight into the anatomical location where L-selectin downregulation occurs in vivo we mapped MDSC expansion and L-selectin density on naïve T cell subsets in lymphoid organs where expression of this LN homing receptor is known to be under tight control. In this regard, during normal development T cell precursors leave the bone marrow and emigrate to the thymus where they

differentiate into L-selectin[+] mature T cells (*Takahama, 2006*). After exiting the thymus, naïve CD4[+] and CD8[+] T cells use L-selectin to traffic directly into LN via gatekeeper HEV, or recirculate through the spleen which is devoid of HEV and does not require L-selectin for entry (*Girard et al., 2012*; *Evans et al., 2015*). In order to maximize the potential for detecting MDSC functions in various lymphoid organs we opted to use the 4T1 mammary tumor model in which tumor-produced granulocyte-colony stimulating factor drives robust expansion of MDSC (*Waight et al., 2011*, *2013*). We excluded non-lymphoid organs and tumor tissues from this analysis since naïve T cells do not recirculate at a high frequency at these sites (*Chen et al., 2006*; *Fisher et al., 2011*).

We found the thymus was devoid of myeloid cells co-expressing the canonical murine MDSC markers CD11b and Gr-1 despite high systemic MDSC burdens in 4T1 tumor-bearing BALB/c female mice (*Figure 1A*). Moreover, L-selectin was not altered on thymic naïve CD4[+]CD44[lo] and CD8[+]-CD44[lo] T cells when compared to non-tumor bearing controls (NTB). MDSC were also rare in peripheral LN (pLN) of 4T1-bearing mice, including tumor-draining inguinal LN, which correlated with the absence of L-selectin modulation on naïve T cells at these sites (*Figure 1A*). MDSC exclusion from LN is likely explained by their low L-selectin expression as compared to normal naïve T cells, (*Figure 1—figure supplement 1*). The uniformly high L-selectin density detected on intranodal T cells in both control and tumor-bearing mice is suggestive of a stringent requirement for a high L-selectin threshold for entry of blood-borne T cells across HEV.

In contrast, we detected profound L-selectin downregulation in naïve CD4[+] and CD8[+] T cells that was associated with substantial MDSC elevation in the blood and splenic compartment of female and male 4T1-bearing mice (*Figure 1A* and *Figure 1—figure supplement 2A*). Splenic CD11b[+]Gr-1[+] cells from 4T1-bearing mice were confirmed to exhibit prototypical MDSC suppressor function defined by potent inhibition of CD3/CD28-driven proliferation of CD4[+] and CD8[+] T cells (from non-tumor bearing mice) in vitro (*Figure 1B*). Conversely, control CD11b[+] cells from non-tumor bearing mice were not immunosuppressive. Data showing an overall decrease in naïve T cells in LN, together with an increase in spleens of 4T1-bearing mice without changes in the apoptotic index (*Figure 1C*), support the notion that suboptimal L-selectin expression reduces T cell access to LN and causes compensatory redistribution to the spleen.

We further determined that L-selectin loss on splenic CD4[+] and CD8[+] T cells correlated temporally with the extent of MDSC expansion which varied among different murine tumor types. Thus, while MDSC expansion early during tumor progression (i.e., 7 days post-4T1 implantation) coincided with significant L-selectin downregulation on naïve CD4[+] and CD8[+] T cell subsets, even greater L-selectin loss occurred at later time-points with higher 4T1 tumor burdens at subcutaneous sites or the mammary fat pad (*Figure 1—figure supplement 2B–D*). Compared to the 4T1 system, MDSC expansion was delayed in C57BL/6 mice implanted with AT-3 mammary tumor cells derived from genetically-engineered MMTV-PyMT/B6 transgenic mice (MTAG) (*Waight et al., 2013*), corresponding with moderate but significant L-selectin downregulation on naïve T cell subsets at ≥21 days post-tumor implantation (*Figure 1—figure supplement 3A and B*). L-selectin downregulation also occurred on splenic T cells in other tumor models including B16 melanoma and CT26 colorectal tumor, but only in rare individual mice with abundant MDSC (*Figure 1—figure supplement 4*), consistent with observations that these tumors do not typically induce MDSC expansion (*Youn et al., 2008*; *Fisher et al., 2011*; *Ito et al., 2015*).

Clues about the spatial regulation of L-selectin emerged from immunohistological staining of the spleen that revealed dense focal accumulations of Gr-1[+] cells congregating with CD3[+] T cells in the marginal zone that were sharply segregated from intrafollicular B220[+] B cells in 4T1-bearing mice (*Figure 1D*). Additional insight into the MDSC mechanism of action came from profiling of splenic B cells that showed that while these cells express relatively low L-selectin compared to T cells as reported previously (*Tang et al., 1998*; *Gauguet et al., 2004*), there was no change in L-selectin density on splenic B cells of tumor-bearing mice (*Figure 1E*). This differential regulation of L-selectin in splenic T and B cells suggests a model in which close physical contact with MDSC is a prerequisite for L-selectin loss.

## L-selectin downregulation occurs on both T and B cells in the blood compartment

To determine if L-selectin loss in T cells is restricted to the spleen or also occurs outside organized lymphoid organs, we performed proof-of-principle experiments in splenectomized mice. Mice were

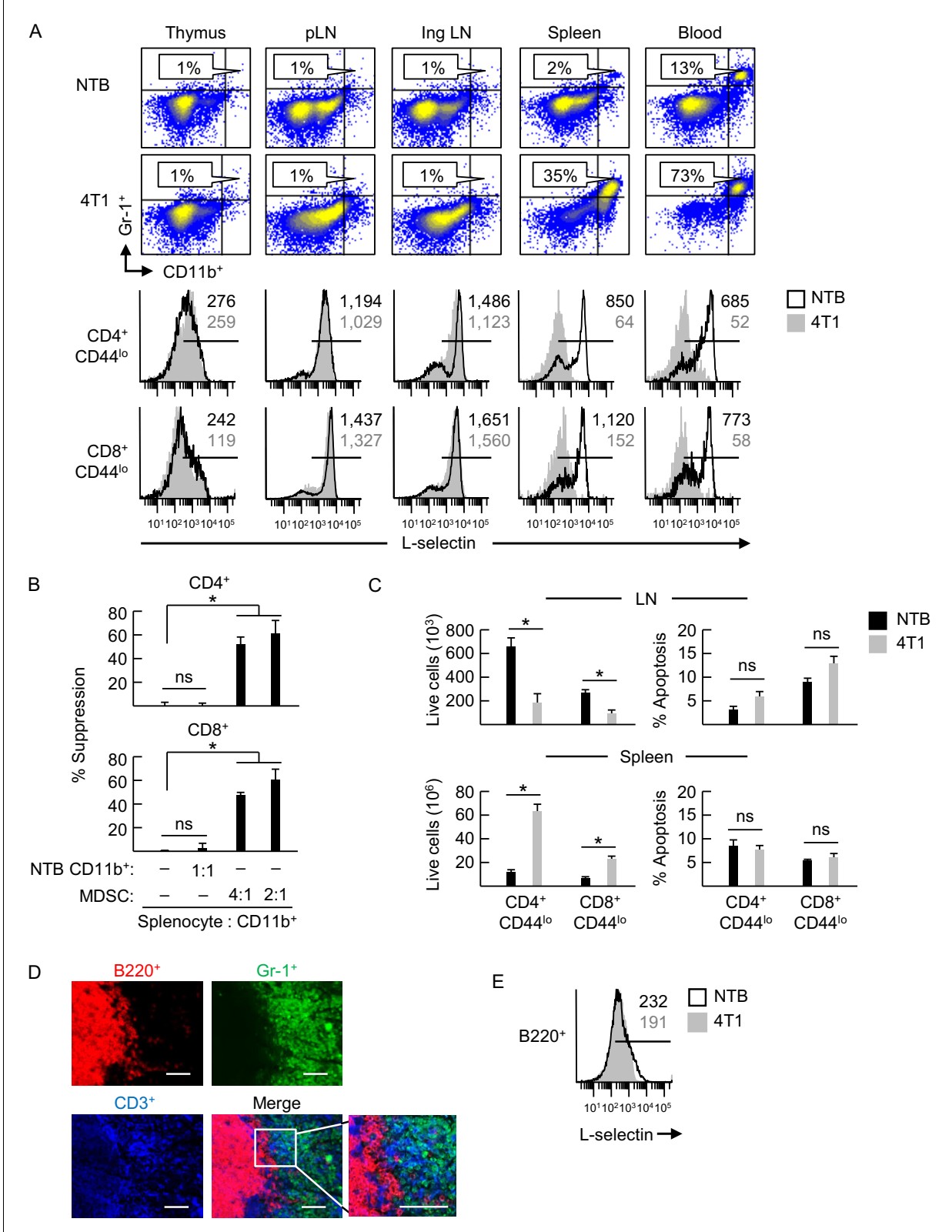

**Figure 1.** L-selectin downregulation on naïve T cells is restricted to specific anatomical compartments associated with MDSC accumulation. (**A**) Flow cytometric analysis demonstrating CD11b⁺Gr-1⁺ MDSC accumulation (% CD45⁺ leukocytes, *top*) and L-selectin expression on CD4⁺CD44$^{lo}$ and CD8⁺CD44$^{lo}$ naïve T cells (*below*) in the indicated lymphoid organs (thymus, peripheral lymph node (pLN), inguinal (Ing) LN, tumor-draining Ing LN, spleen, and blood) of non-tumor bearing (NTB) mice or 4T1-bearing mice (tumor volume 1150 ± 150 mm³). (**B**) MDSC-T cell suppression assay. Splenic

*Figure 1 continued on next page*

*Figure 1 continued*

CD11b[+] cells either from NTB mice or 4T1-bearing mice (tumor volume 2600 ± 380 mm[3]) were co-cultured with CFSE-labeled target splenocytes from NTB mice at the indicated splenocyte:myeloid cell ratios. Proliferation (based on CFSE dilution) in T cell subsets was measured 72 hr after addition of anti-CD3/CD28 antibody-conjugated activation beads. Percent suppression is for one experiment (mean±s.e.m, *n* = 3 replicates per condition) and is representative of three independent experiments. (C) Total numbers of viable naïve CD4[+]CD44[lo] and CD8[+]CD44[lo] T cell subsets (*left*) and percentages of annexin V[+] early apoptotic T cells (*right*) were quantified by flow cytometric analysis from peripheral lymph nodes and spleens of NTB or 4T1-bearing mice (tumor volume 1340 ± 242 mm[3]). Data (mean±s.e.m.) are of one experiment (*n* = 3 mice per group) and are representative of two independent experiments. (B,C) *p<0.05; ns, not significant; data were analyzed by unpaired two-tailed Student's *t*-test. (D) Splenic cryosections from NTB and 4T1-bearing mice stained for B220[+], Gr-1[+] and CD3[+] cells; parallel fluorocytometric analysis (as in A) confirmed that >90% of splenic Gr-1[+] cells co-expressed CD11b. Scale bar, 50 μm. (E) L-selectin expression on splenic B220[+] cells of NTB and 4T1–bearing mice. (A,E) Horizontal lines in histograms indicate positively stained cells; numbers are mean fluorescence intensity. (A,B,D–E) Data are for one experiment and are representative of ≥ three independent experiments (*n* = 3 replicates or mice per group). pLN, peripheral lymph node; Ing LN, inguinal lymph node; NTB, non-tumor bearing.

The following figure supplements are available for figure 1:

**Figure supplement 1.** MDSC express low levels of L-selectin.

**Figure supplement 2.** Inverse correlation between MDSC expansion and L-selectin expression on naive CD4[+] and CD8[+] T cells during 4T1 tumor progression.

**Figure supplement 3.** MDSC expansion coincides with L-selectin downregulation on naive CD4[+] and CD8[+] T cells during AT-3 tumor progression.

**Figure supplement 4.** L-selectin down-modulation is associated with MDSC expansion in different tumor types.

sham-surgically treated or splenectomized 10 days prior to implantation of 4T1 tumors (*Figure 2A*). We validated that 4T1 tumor growth and MDSC expansion in the blood was equivalent in sham and splenectomized mice at 21 days post-tumor implantation (*Figure 2B and C*), offsetting any concern that splenectomy might reduce the circulating MDSC burden as reported for lung tumor models (*Cortez-Retamozo et al., 2012*; *Levy et al., 2015*). We then performed adoptive cell transfer (ACT) of L-selectin[hi]CD8[+]CD44[lo] splenic T cells derived from non-tumor bearing mice. L-selectin fate was assessed on transferred cells recovered from the blood after 24 hr.

High L-selectin expression was maintained following transfer of CD8[+] T cells (input cells) into sham, non-tumor bearing controls whereas substantial L-selectin loss occurred within a 24 hr window after transfer into sham-treated 4T1-bearing mice with high MDSC burdens (*Figure 2C*). We surprisingly found that the extent of L-selectin downregulation on adoptively-transferred CD8[+] T cells was indistinguishable whether tumor-bearing recipient mice underwent sham surgery or splenectomy. These results suggested that L-selectin down-modulation on CD8[+] T cells can occur in the intravascular space and does not require structural scaffolds provided by organized tissue compartments.

Additional studies examined L-selectin regulation on endogenous circulating CD4[+] T cells and B220[+] B cells in the context of splenectomy. Like CD8[+] T cells, we found that L-selectin was strongly downregulated on CD4[+] T cells of tumor-bearing mice regardless of whether there was an intact splenic microenvironment (*Figure 2—figure supplement 1*). In the case of B220[+] B cells, the varying levels of baseline L-selectin detected in lymphoid organs of tumor-free mice was expected since L-selectin is known to fluctuate after B cells exit the bone marrow and recirculate through the blood and spleen (*Tang et al., 1998*; *Morrison et al., 2010*). Moreover, we did not detect L-selectin loss in the bone marrow and splenic compartments despite increased MDSC burden in sham or splenectomized tumor-bearing mice (*Figure 3*). In sharp contrast, L-selectin was nearly completely downregulated on blood-borne B220[+] B cells in both sham and splenectomized tumor-bearing mice (*Figure 3*). These observations allowed us to pinpoint the blood compartment as the preferential site of L-selectin modulation for B cells in tumor-bearing mice. Similar results were obtained for L-selectin downregulation on circulating T and B lymphocytes if the timing sequence was reversed by allowing tumor-induced MDSC to accrue prior to splenectomy (*Figure 3—figure supplement 1*). The conclusion that blood is a prominent site of L-selectin loss was further supported by the strong L-selectin downregulation observed only 2 hr after adoptive transfer of naïve CD8[+] and CD4[+] T cells and B220[+] B cells into the MDSC-rich vascular compartment of 4T1-bearing mice (*Figure 3—figure supplement 2*). Labeled cells detected in the blood at this short time-point mainly represent

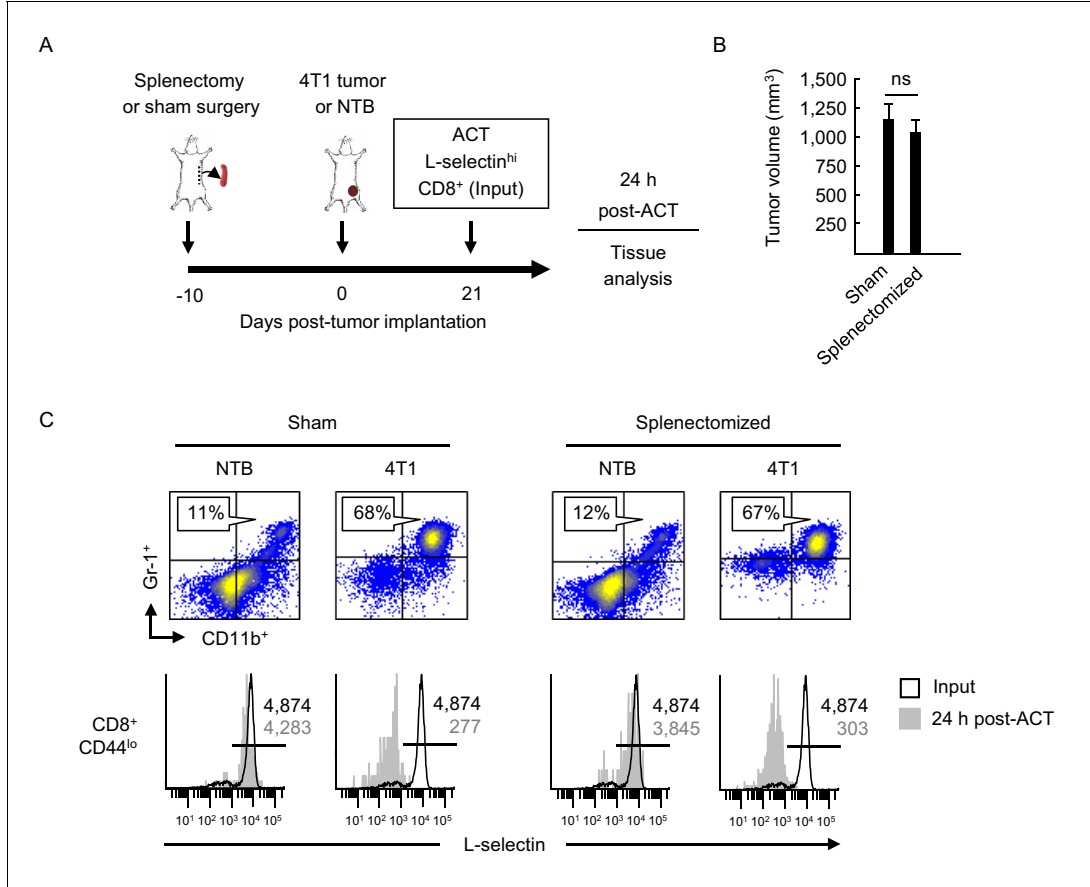

**Figure 2.** L-selectin loss occurs on naïve T cells within the MDSC-enriched blood compartment of splenectomized mice. (**A**) Experimental design in which splenectomy or sham surgery was performed in non-tumor bearing (NTB) mice. Mice were then inoculated with 4T1 tumor or maintained as NTB controls. Fluorescently-labeled L-selectin[hi] CD8[+] T cells isolated from NTB mice (input) were used for intravenous adoptive cell transfer (ACT) into tumor-bearing mice or in NTB controls. (**B**) 4T1 tumor volume of sham and splenectomized mice at 21 days post-4T1 implantation. Data (mean±s.e.m.) are for a single representative experiment ($n$ = 3 mice per group); ns, not significant; data were analyzed by unpaired two-tailed Student's $t$-test. (**C**) Representative flow cytometric analysis showing accumulation of CD11b[+]Gr-1[+] cells (% CD45[+] leukocytes, *top*) and L-selectin expression (*bottom*) on CD8[+]CD44[lo] T cells before ACT (input) and 24 hr post-ACT in the blood of sham or splenectomized NTB and 4T1-bearing recipient mice. Horizontal lines in histograms indicate positively stained cells; numbers are mean fluorescence intensity. (**A–C**) Data are representative of three independent experiments ($n$ = 3 mice per group). NTB, non-tumor bearing; ACT, adoptive cell transfer.

The following figure supplement is available for figure 2:

**Figure supplement 1.** L-selectin downregulation on naïve CD4[+]CD44[lo] T cells occurs in the MDSC-enriched peripheral blood compartment of splenectomized mice.

transferred populations retained in the vascular compartment since 2 hr is not sufficient for lymphocytes to recirculate from blood, through tissues, and back to the blood (e.g., transit times for T cells through lymph nodes, spleen, and peripheral tissues are ~8–12, 5, and 24 hr, respectively, and ~24 hr for B cells at these tissue sites) (*Ford, 1979*; *Issekutz et al., 1982*; *Girard et al., 2012*).

L-selectin loss was also detected in circulating T and B cells in MTAG mice with a high cumulative mammary tumor burden (~9000 mm[3]) and moderate MDSC expansion (14% of CD45[+] peripheral blood cells), but not in MTAG mice with low MDSC (7% of CD45[+] cells) and tumor burdens (~2500 mm[3]) (*Figure 3—figure supplement 3*). L-selectin on CD3[+]CD45RA[+] naïve human T cells was additionally shown to be subject to downregulation following adoptive transfer of normal donor-derived human peripheral blood lymphocytes into MDSC[hi] 4T1-bearing severe-combined immunodeficient (SCID) mice (*Figure 3—figure supplement 4*), indicating that a non-species restricted mechanism was operative in vivo. Collectively, these findings provide evidence that the MDSC-enriched blood

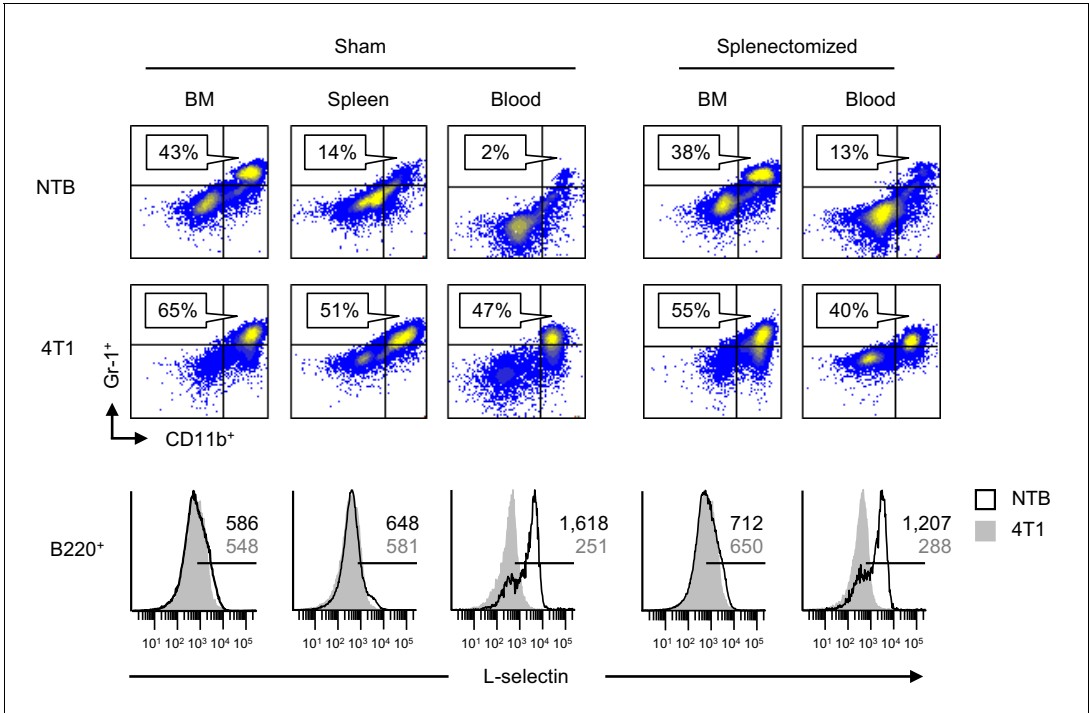

**Figure 3.** L-selectin downregulation on B cells occurs exclusively in the peripheral blood. Splenectomy or sham surgery was performed 10 days prior to 4T1 tumor inoculation and tissues were evaluated for MDSC expansion and L-selectin expression 22 days after tumor implantation (tumor volume ~1000 mm³). Flow cytometric analysis of CD11b⁺Gr-1⁺ cell burden (% CD45⁺ leukocytes; *top*) and L-selectin expression on endogenous B220⁺ B cell populations (*bottom*) in the indicated organs (BM, bone marrow; spleen; blood) of sham or splenectomized non-tumor bearing (NTB) and 4T1-tumor bearing mice. Horizontal lines on histograms indicate positively stained cells; numbers are mean fluorescence intensity. Data are representative of ≥ three independent experiments (*n* = 3 mice per group); NTB, non-tumor bearing; BM, bone marrow.

The following figure supplements are available for figure 3:

**Figure supplement 1.** L-selectin downregulation on T and B cells occurs in the MDSC-enriched peripheral blood compartment of splenectomized mice.

**Figure supplement 2.** L-selectin loss on naïve T and B lymphocytes occurs rapidly in the blood of tumor-bearing mice.

**Figure supplement 3.** MDSC-associated downregulation of L-selectin on naïve T and B lymphocytes occurs in autochthonous mammary carcinoma MTAG mice.

**Figure supplement 4.** L-selectin on human naïve T cells is downregulated following transfer into 4T1 tumor-bearing mice.

compartment of tumor-bearing mice is a major site of L-selectin downregulation and that both T and B lymphocytes are targeted.

## MDSC cause L-selectin loss through a contact-mediated mechanism independent of ADAM17

MDSC were implicated in regulating L-selectin expression on T cells in tumor-bearing mice in prior studies using gemcitabine cytotoxic chemotherapy (*Hanson et al., 2009*) which kills MDSC in vitro and in vivo (*Vincent et al., 2010*; *Mundy-Bosse et al., 2011*). However, since gemcitabine has a wide spectrum of activities that also influence other immune cells (e.g., CD4⁺ Treg and Th17 cells) (*Bracci et al., 2014*), we took several alternative approaches to further explore MDSC contributions to reducing L-selectin in vivo. Partial MDSC depletion (~50%) by administration of anti-Gr-1 antibody at three day intervals after 4T1 implantation had no impact on tumor growth but significantly rescued L-selectin on circulating naïve T and B cells (*Figure 4A and B*). Complementary studies showed

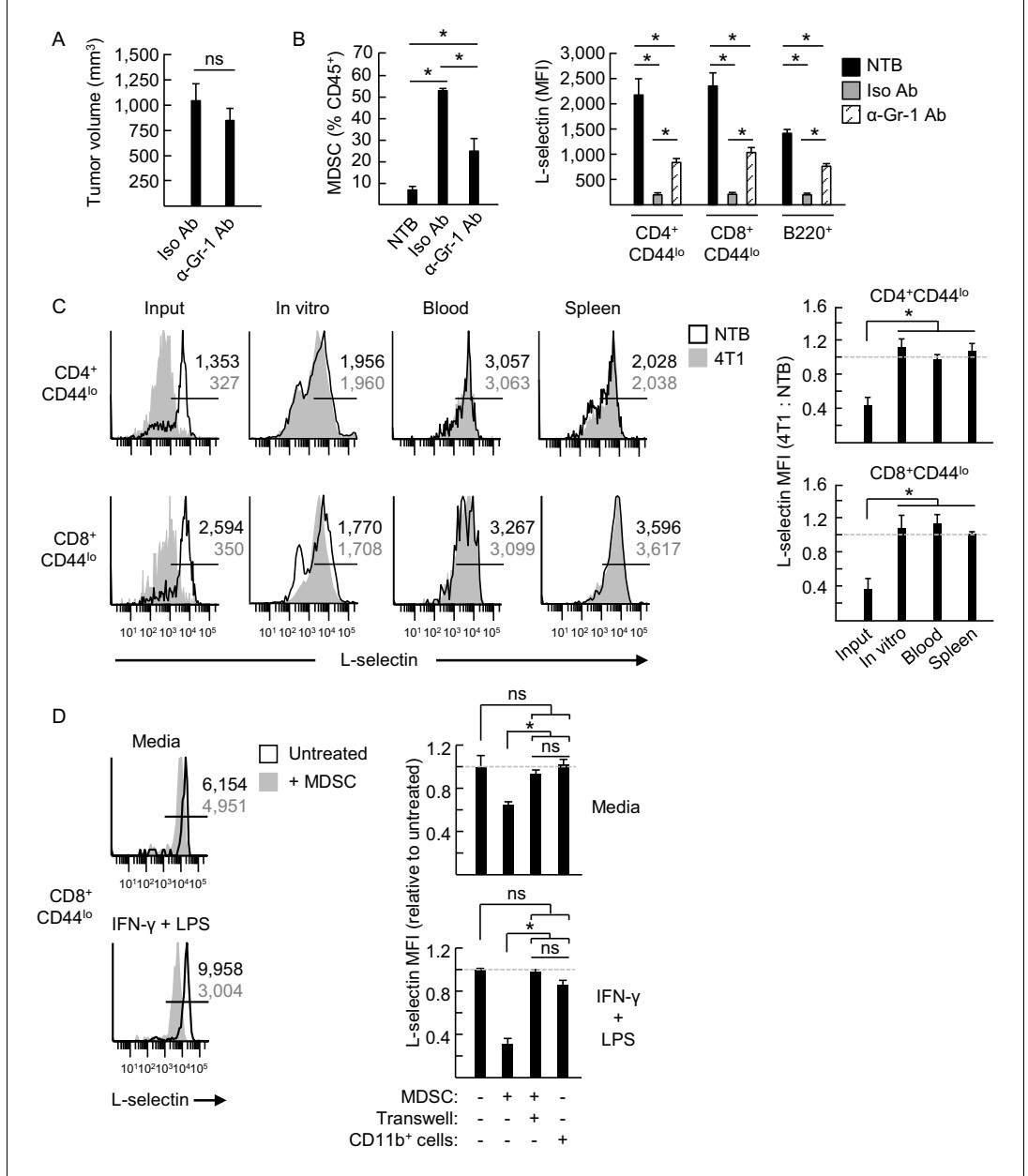

**Figure 4.** MDSC induce L-selectin loss on T and B lymphocytes in 4T1 tumor-bearing mice via a contact-dependent mechanism. (**A**) 4T1-tumor-bearing mice were treated with anti-Gr-1 antibodies (α-Gr-1 Ab) or isotype control antibodies (Iso Ab) every 3 days for three weeks starting at three days post-tumor implantation. Endpoint tumor volumes are shown. (**B**) CD11b⁺Gr-1⁺ MDSC burden (% CD45⁺ leukocytes, *left*) and L-selectin expression (mean fluorescence intensity, MFI) on endogenous CD4⁺CD44ˡᵒ, CD8⁺CD44ˡᵒ, and B220⁺ lymphocytes (*right*) were measured in the blood of NTB or in 4T1-bearing mice treated with Iso Ab or anti-Gr-1 Ab. (**C**) Splenocytes from NTB or 4T1-bearing mice were depleted of CD11b⁺ cells by magnetic bead isolation (94.8 ± 1.8% depletion, *n* = 3 mice). These cell populations were then fluorescently-labeled with different tracking dyes, co-mixed at a 1:1 ratio, and cultured in vitro or adoptively transferred into NTB recipients. Representative flow cytometric L-selectin profiles are shown for naïve CD4⁺CD44ˡᵒ and CD8⁺CD44ˡᵒ T cells before culture or adoptive cell transfer (ACT) (input) and four days after in culture (in vitro) or for cells recovered from blood and spleen post-ACT (*left*). Quantification of L-selectin modulation (*right*) is based on a ratio of the MFI for T cells from 4T1-bearing mice relative to NTB mice; dashed lines indicate NTB control. (**D**) MDSC or CD11b⁺ control cells were isolated from 4T1-tumor bearing mice (tumor volume >1000 mm³) or NTB mice, respectively. Myeloid cells were then co-cultured with fluorescently-labeled splenocytes from NTB mice (10:1 ratio) in media alone or with IFN-γ (20 U/mL) and LPS (100 ng/mL). MDSC and splenocytes were separated by transwell inserts (0.4 μm pore size) in the indicated co-cultures. After 24 hr, L-selectin expression on viable naive CD8⁺CD44ˡᵒ T cells was analyzed by flow cytometry; representative profiles are shown (*left*). Relative changes in L-selectin expression were normalized to untreated CD8⁺CD44ˡᵒ T cells (indicated by dashed lines; *right*). (**A–D**) Data (mean±s.e.m.) are from one experiment (*n* = 3 mice per group or ≥3 replicates per group) and are representative of ≥ two independent experiments. *p<0.05; ns, not

*Figure 4 continued on next page*

Figure 4 continued

significant; data were analyzed by unpaired two-tailed Student's *t*-test. (C,D) Horizontal lines in histograms indicate positively stained cells; numbers are MFI. NTB, non-tumor bearing; Ab, antibody; Iso, isotype; MFI, mean fluorescence intensity.

that L-selectin loss was reversible in an environment devoid of MDSC. In this regard, when CD11b[+] MDSC were depleted from L-selectin[lo] splenic populations from 4T1-bearing mice we observed complete L-selectin recovery on CD4[+] and CD8[+] T cells within four days after culture or post-adoptive transfer into non-tumor bearing recipient mice (*Figure 4C*). Thus, these findings demonstrate that continued exposure to MDSC is required to maintain L-selectin down-modulation. Finally, MDSC from tumor-bearing mice (>95% CD11b[+]Gr-1[+]) but not CD11b[+] cells from non-tumor bearing controls were shown to cause moderate but significant L-selectin loss during co-culture with splenic CD8[+] T cells for 24 hr (*Figure 4D*). Stronger L-selectin loss occurred if IFN-γ and LPS were included in co-cultures to sustain MDSC function ex vivo (*Sinha et al., 2007*; *Stewart et al., 2009*; *Ostrand-Rosenberg et al., 2012*; *Beury et al., 2014*). Moreover, MDSC acted through a contact-dependent mechanism that was abrogated if MDSC were physically separated from target lymphocytes by cell-impermeable transwell inserts (*Figure 4D*). Taken together, these results establish that MDSC directly target lymphocytes for L-selectin loss.

Further investigation into the mechanisms underlying MDSC activity showed that L-selectin mRNA levels detected by quantitative RT-PCR were unchanged in splenic CD4[+] and CD8[+] T cells of 4T1-bearing mice compared to non-tumor bearing controls (*Figure 5A*), indicating that L-selectin loss does not involve transcriptional repression in vivo. MDSC-mediated L-selectin downmodulation was instead accompanied by a 2.5-fold increase in soluble (s)L-selectin in the serum which was in line with a sheddase-dependent mechanism operative in vivo (*Figure 5B*). L-selectin is a well-known target of the ADAM17 ecto-protease which operates in *cis* to cleave substrates on the same membrane surface (*Feehan et al., 1996*). However, reports that MDSC express surface ADAM17 (*Hanson et al., 2009*; *Oh et al., 2013*; *Parker et al., 2014*) have raised the possibility of a non-conventional *trans*-acting mechanism whereby MDSC-intrinsic ADAM17 cleaves L-selectin on juxtaposed T cells. Thus, we were prompted to systematically investigate the role of ADAM17 in MDSC-induced L-selectin loss.

Head-to-head comparison between the well-established phorbol myristate acetate (PMA)-induced ADAM17 pathway versus MDSC-directed L-selectin loss on the surface of CD8[+] T cells (*Figure 5C and D*) or CD4[+] T cells and B220[+] B cells (*data not shown*) revealed a sharp demarcation in their ADAM17 requirements in vitro. Thus, in agreement with the obligate role of ADAM17 reported for PMA-induced L-selectin shedding, we found that PMA-induced loss of lymphocyte L-selectin was abrogated (*1*) by an ADAM17-specific inhibitor (PF-5480090) (*McGowan et al., 2013*) and by a dual ADAM17/ADAM10 inhibitor (INCB7839) (*Witters et al., 2008*; *Wang et al., 2013*); (*2*) in cells from L(E)-selectin mice expressing a mutated ADAM cleavage site due to substitution of the L-selectin membrane-proximal extracellular domain with the shorter E-selectin homologous domain (*Venturi et al., 2003*); or (*3*) in ADAM17-deficient lymphocytes from *Adam17[flox/flox]/Vav1-Cre* mice (*Mishra et al., 2016*) cultured alone (*Figure 5C*) or co-mixed with wildtype cells (*Figure 5—figure supplement 1*). These findings confirm reports of a strict requirement for *cis*-acting ADAM17 for PMA-induced L-selectin down-modulation (*Feehan et al., 1996*; *Preece et al., 1996*). In contrast, MDSC-induced L-selectin downregulation in vitro was unaffected by inhibitors of ADAM17 or ADAM17/ADAM10; L(E)-selectin mutation; or lymphocyte-intrinsic ADAM17 deficiency (*Figure 5D*). Further, while elevated constitutive L-selectin expression in mutant L(E)-selectin lymphocytes or on *Adam17[−/−]* cells was indicative of an ADAM17 mechanism operative in vivo as described previously (*Venturi et al., 2003*; *Li et al., 2006*), this pathway was dispensable for MDSC-induced L-selectin downregulation in mutant L(E)-selectin-expressing T and B cells or in *Adam17[−/−]* cells following their adoptive transfer into MDSC[hi] 4T1-bearing SCID mice (*Figure 5E*). Collectively, these data exclude a role for ADAM17 or ADAM10 in either a *cis* or *trans* orientation for MDSC-induced L-selectin loss and are suggestive of the involvement of another ecto-protease.

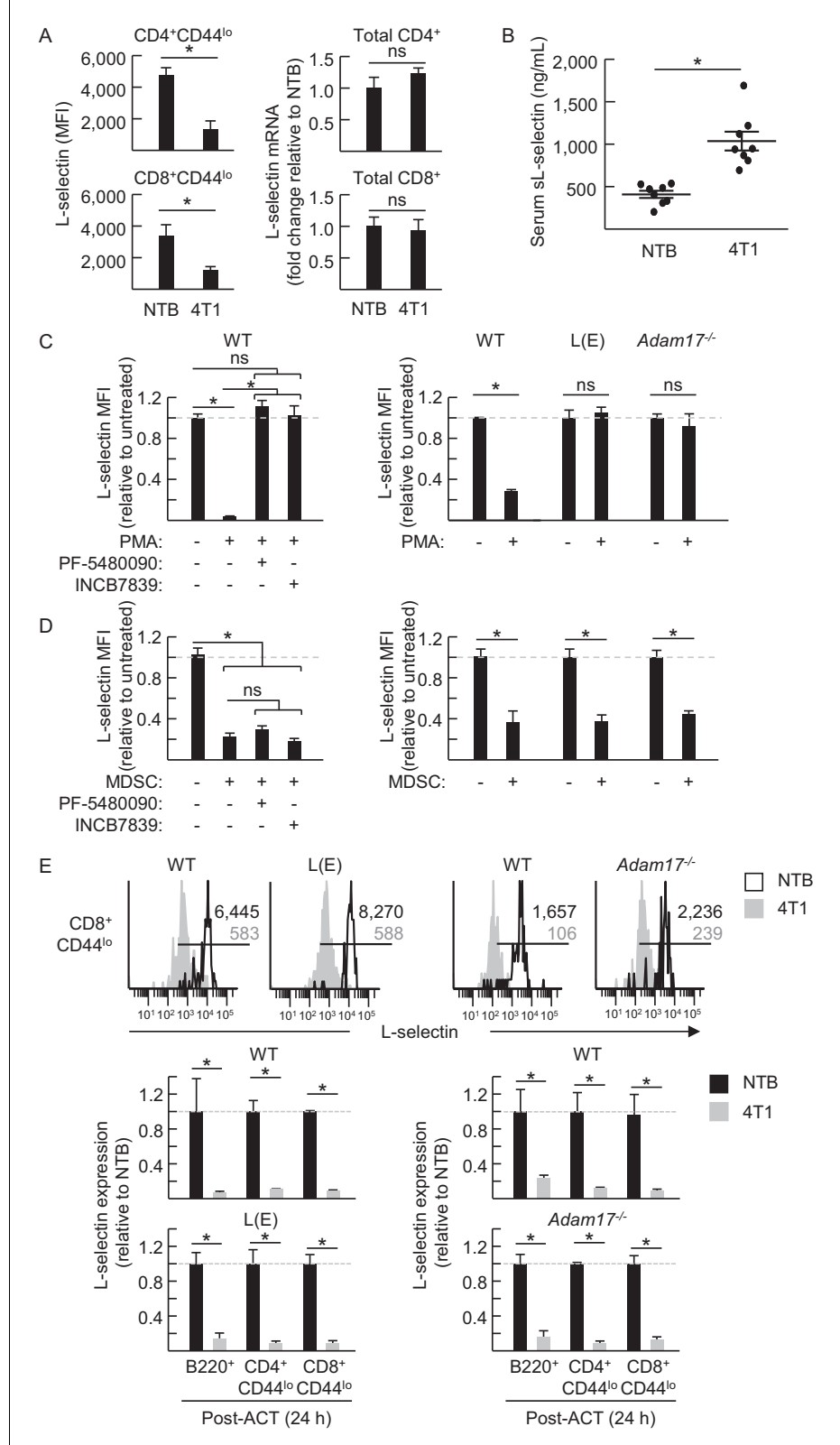

**Figure 5.** MDSC-induced L-selectin downregulation is post-transcriptional and does not depend on the ADAM17 metalloprotease. (**A**) Flow cytometric analysis of surface L-selectin (mean fluorescence intensity; MFI) on splenic naïve CD4+CD44lo and CD8+CD44lo T cells of non-tumor bearing (NTB) and 4T1-bearing mice is shown (*left*). L-selectin mRNA expression in splenic CD4+ and CD8+ T cells from NTB and 4T1-bearing mice was determined by

*Figure 5 continued on next page*

*Figure 5 continued*

qRT-PCR with fold-change normalized with β-actin (*right*). (**B**) Soluble (s)L-selectin in serum of individual NTB and 4T1-bearing mice was assessed by ELISA. (**A,B**) Data are from three independent experiments ($n \geq 2$ mice per group in each experiment; tumor volume >1500 mm³; average frequency of splenic CD11b⁺Gr-1⁺ cells (% CD45⁺ leukocytes) in tumor-bearing mice was ~40%). (**C**) Splenocytes were isolated from NTB wildtype (WT) C57BL/6 mice, L(E)-selectin transgenic mice, $Adam17^{flox/flox}$/Vav1-Cre mice ($Adam17^{-/-}$), or age-matched WT littermate controls. WT splenocytes were pretreated for 30 min with the ADAM17-specific inhibitor PF-5480090 (10 µM) or the ADAM17/10-specific inhibitor INCB7839 (20 µM). WT, L(E), and $Adam17^{-/-}$ splenocytes were then cultured 2 hr with or without phorbol myristate acetate (PMA, 100 ng/mL). Surface L-selectin (MFI) on viable naïve CD8⁺CD44ˡᵒ T cells relative to untreated controls was determined by fluorocytometric analysis. (**D**) Splenic CD11b⁺Gr-1⁺ MDSC were purified from 4T1-bearing mice; splenocytes were from various NTB mice as described in (**C**). MDSC and WT splenocytes were both pretreated for 30 min with or without PF-5480090 or INC7839. MDSC and splenocytes from WT, L(E), or $Adam17^{-/-}$ mice were then co-cultured at a 10:1 ratio for 24 hr in media containing IFN-γ (20 U/mL) and LPS (100 ng/mL). L-selectin on viable naïve CD8⁺CD44ˡᵒ T cells was assessed by flow cytometric analysis. (**E**) Fluorescently-labeled WT, L(E), and $Adam17^{-/-}$ splenocytes (i.e., from NTB mice) were adoptively transferred into NTB severe-combined immunodeficient (SCID) mice or 4T1-bearing SCID mice at 21 days-post tumor implantation (average tumor volume for all experiments, 1102 ± 191 mm³; average circulating CD11b⁺Gr-1⁺ frequencies in NTB SCID recipients, 75 ± 8 cells/µL blood, and 4T1-bearing SCID recipients, 4081 ± 876 cells/µL blood). After 24 hr post-ACT, L-selectin was assessed by flow cytometry on transferred splenocytes recovered from the blood of NTB and 4T1-bearing SCID mice. Representative flow histograms depict L-selectin expression on naïve CD8⁺CD44ˡᵒ T cells (*above*); horizontal lines indicate positively stained cells, numbers are mean fluorescence intensity. Normalized data for L-selectin expression on B220⁺, CD4⁺CD44ˡᵒ, CD8⁺CD44ˡᵒ cells 24 hr post-adoptive transfer are for one representative experiment ($n \geq 2$ mice per group) (*below*). (**A–E**) *$p<0.05$; ns, not significant; all data (mean±s.e.m.) were analyzed by unpaired two-tailed Student's *t*-test. (**C–E**) Data are representative of $\geq$ two independent experiments ($n \geq 2$ replicates or mice per group) and are normalized to untreated or NTB controls (indicated by dashed lines). NTB, non-tumor bearing; WT, wildtype; MFI, mean fluorescence intensity; sL-selectin, soluble L-selectin; ACT, adoptive cell transfer.

The following figure supplement is available for figure 5:

**Figure supplement 1.** PMA-induced loss of L-selectin depends on *cis*-acting ADAM17.

## L-selectin loss reduces murine CD8⁺ T cell trafficking across LN HEV

Observations that early tumor development is associated with moderate L-selectin loss raised the question of whether this would be sufficient to compromise trafficking, particularly since L-selectin is present in such excess on leukocyte surface membranes (*Kishimoto et al., 1989*; *Simon et al., 1992*). To address the functional consequence of moderate L-selectin loss we isolated L-selectinʰⁱ CD8⁺ T cells (>90% purity) from spleens of non-tumor bearing controls (NTB CD8⁺) or L-selectin intermediate-to-low (L-selectinⁱⁿᵗ/ˡᵒ) CD8⁺ cells from AT-3-bearing mice (AT-3 CD8⁺) (*Figure 6A*). Cells were then labeled ex vivo with tracking dye and their adhesive behavior was visualized in real-time by epifluorescence intravital microscopy in LN venules of non-tumor bearing recipients (*Chen et al., 2006*). For L-selectinʰⁱ CD8⁺ T cells, tethering and rolling interactions and firm sticking occurred primarily in high-order (III-V) postcapillary venules that constitute the HEV (*Figure 6B and C*; *Video 1*) (*Chen et al., 2006*). L-selectin-mediated tethering and slow rolling on HEV ligands termed peripheral LN addressin (PNAd) is a prerequisite for CC-chemokine receptor-7 (CCR7) engagement of CCL21 which, in turn, triggers stable binding of LFA-1 integrin to endothelial ICAM-1/2 (*Girard et al., 2012*; *Evans et al., 2015*). As expected, minimal adhesion of L-selectinʰⁱ CD8⁺ T cells occurred in low-order venules (LOV; e.g., order II venules) (*Figure 6B and C*) that lack PNAd (*M'Rini et al., 2003*) and CCL21 (*Stein et al., 2000*).

L-selectinⁱⁿᵗ/ˡᵒCD8⁺ T cells from AT-3-bearing mice surprisingly exhibited a normal frequency of tethering and rolling interactions in HEV (*Figure 6C*; *Video 2*). However, we detected a significant decrease in their transition from rolling to firm arrest in order IV and V venules (*Figure 6C*; *Video 2*) despite normal expression and function of CCR7 and LFA-1, as determined by flow cytometric profiling, CCL21-driven chemotaxis assays, and LFA-1–dependent homotypic aggregation assays (*Figure 6—figure supplement 1A–C*). An explanation for these paradoxical findings was revealed by data showing defective rolling behavior of L-selectinⁱⁿᵗ/ˡᵒ CD8⁺ T cells, as evidenced by their

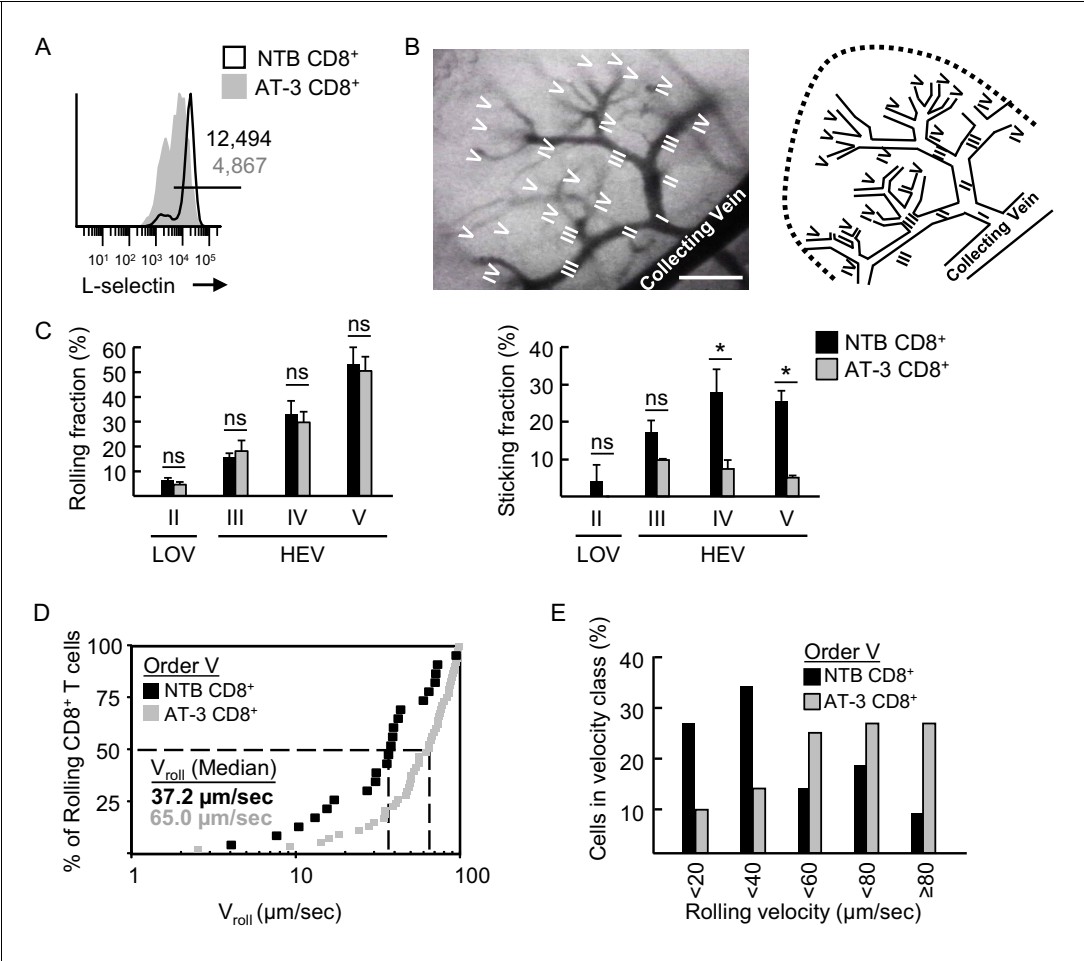

**Figure 6.** L-selectin-deficient CD8[+] T cells from AT-3-bearing mice exhibit reduced firm adhesion and faster rolling velocity in LN HEV. (**A**) Flow cytometric analysis of L-selectin expression prior to intravenous adoptive transfer of CD8[+] T cells from non-tumor bearing mice (NTB CD8[+]) or CD8[+] T cells from AT-3–bearing mice (AT-3 CD8[+]). Horizontal line in histogram indicates positively stained cells; numbers are mean fluorescence intensity. (**B**) Representative photomicrograph (*left*) and schematic (*right*) of the postcapillary vascular tree visualized by epifluorescence intravital microscopy in inguinal LN of NTB recipient mice. Hierarchical branches of venular orders I and II (low-order venules, LOV) and III-V (high-order venules corresponding to HEV) are labeled. Direction of blood flow in post-capillary venules is from order V to order I venules which directly empty into collecting veins. Scale bar, 100 μm. (**C**) Rolling fraction and sticking fraction of fluorescently-labeled CD8[+] T cells isolated from NTB or AT-3–bearing mice following adoptive transfer into NTB recipients. Data (mean±s.e.m.) are from three independent experiments; $n \geq 3$ mice per group. *p<0.05; ns, not significant; data were analyzed by unpaired two-tailed Student's *t*-test. (**D**) Cumulative rolling velocity curve was generated by measuring the velocities of transferred CD8[+] T cells in order V venules of inguinal LN in three independent experiments. Comparison of cumulative rolling velocity plot data was performed by unpaired two-tailed Student's *t*-test; L-selectin[hi] NTB CD8[+] T cells versus L-selectin[int/lo] AT-3 CD8[+] T cells, *p<0.01. (**E**) Distributions of rolling velocities in velocity histograms in order V venules were evaluated by a nonparametric Mann-Whitney U test; L-selectin[+] NTB CD8[+] T cells versus L-selectin[int/lo] AT-3 CD8[+] T cells, *p<0.01. NTB, non-tumor bearing.

The following figure supplement is available for figure 6:

**Figure supplement 1.** Adhesion molecule expression and function on CD4[+] and CD8[+] T lymphocytes.

significantly faster median rolling velocity when compared to L-selectin[hi] CD8[+] T cells in order V venular segments (65.0 versus 37.2 μm/sec, respectively; *Figure 6D and E*).

Competitive short-term homing assays further identified a defect in the ability of L-selectin[int/lo] cells to extravasate across LN HEV and enter the underlying parenchyma. For these studies, enriched populations of L-selectin[hi] and L-selectin[int/lo] CD8[+] T cells from non-tumor bearing mice and AT-3–bearing mice, respectively, were labeled with different tracking dyes ex vivo, co-mixed at a 1:1 ratio, and transferred intravenously into tumor-free recipients. The impact of L-selectin deficits on

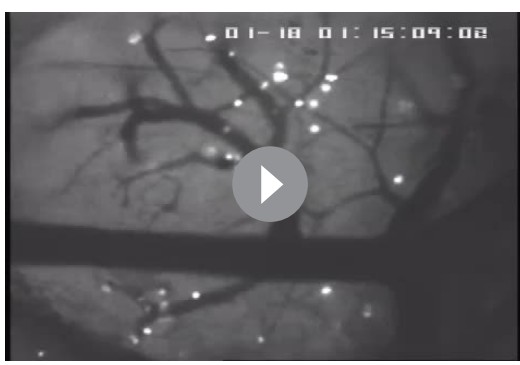

**Video 1.** Real-time intravital imaging of L-selectin[hi] CD8[+] T cells trafficking in lymph node HEV. Intravital imaging of fluorescently-labeled L-selectin[hi] CD8[+] T cells from a non-tumor bearing mouse undergoing transient rolling interactions and firm arrest within postcapillary venules of an inguinal lymph node in a non-tumor bearing mouse. The initial injection of transferred CD8[+] cells occurred ~12 min prior to capture of images.

trafficking was assessed 1 hr later in peripheral LN under homeostatic (control) conditions and under conditions of heightened HEV function as encountered in inflamed lymph nodes (*Evans et al., 2015*). Homeostatic trafficking of L-selectin[int/lo] CD8[+] T cells across LN HEV was strongly inhibited (~80%) when compared to L-selectin[hi] T cells (*Figure 7A*). Transferred CD8[+] T cells that successfully extravasated across LN HEV during homeostatic trafficking were uniformly L-selectin[hi] regardless of whether they originated from non-tumor bearing mice or AT-3–bearing mice (*Figure 7B*), suggesting that a high L-selectin density is necessary to stabilize T cell adhesion during extravasation. In contrast, differential L-selectin expression was maintained on transferred CD8[+] T cells recovered from the spleen where trafficking is not dictated by L-selectin status (*Figure 7A and B*). Reduced CD8[+] T cell trafficking due to moderate L-selectin loss was also observed in an inflammatory model in which HEV express elevated CCL21 and ICAM-1 in response to fever-range whole body hyperthermia (WBH) (*Figure 7A*; *Figure 7–figure supplement 1*) (*Chen et al., 2006*; *Evans et al., 2015*). Taken together, these results establish the biological significance of MDSC-induced L-selectin loss in limiting T cell access to LN via HEV portals.

## Reduced L-selectin-dependent trafficking compromises antigen-driven activation in LN

We formally tested the prediction that MDSC-directed downregulation of L-selectin-dependent trafficking diminishes T cell responses to cognate antigen within the LN compartment using CD8[+] OT-I transgenic mice expressing T cell receptors (TcR) specific for ovalbumin residues 257-264 (OVA$_{257-264}$; SIINFEKL). L-selectin[hi] and L-selectin[int/lo] CD8[+] T cells were purified from spleens of non-tumor bearing OT-I mice and AT-3-bearing OT-I mice, respectively (e.g., as in *Figure 6A*). These OT-I cells were then labeled ex vivo with different proliferation dyes and co-mixed at a 1:1 ratio to assess functional responses to antigen in competitive activation assays in vitro and in vivo (*Figure 8A*).

Since prior studies reported that T cells isolated from tumor-bearing mice and cancer patients have intrinsically diminished antigen responsiveness (*Alexander et al., 1993*; *Jiang et al., 2015*), we first established a relative baseline level of function for CD8[+] OT-I T cells from tumor-bearing mice under in vitro conditions where access to antigen is L-selectin–independent (*Figure 8A and B*). Co-cultures of OT-I cells from non-tumor bearing mice and AT-3–bearing transgenic mice (i.e., 1:1 ratio) were stimulated in vitro for four days with SIINFEKL-loaded bone marrow-derived dendritic cells (DC). OT-I T cells from non-tumor bearing mice were ≥2 times more responsive to antigen-

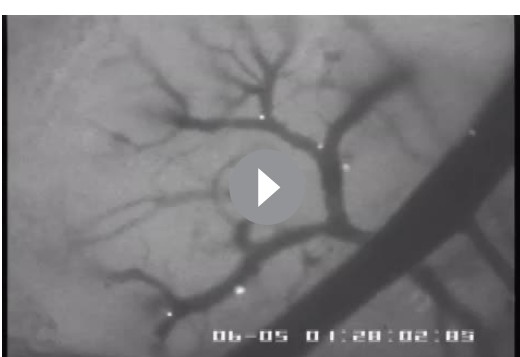

**Video 2.** Intravital imaging of L-selectin[lo] CD8[+] T cells trafficking in lymph node HEV. Impaired sticking of calcein-labeled L-selectin[lo] CD8[+] T cells from an AT-3-bearing mouse within postcapillary venules of an inguinal lymph node in a non-tumor bearing mouse. The initial injection of transferred CD8[+] cells occurred ~13 min prior to capture of images.

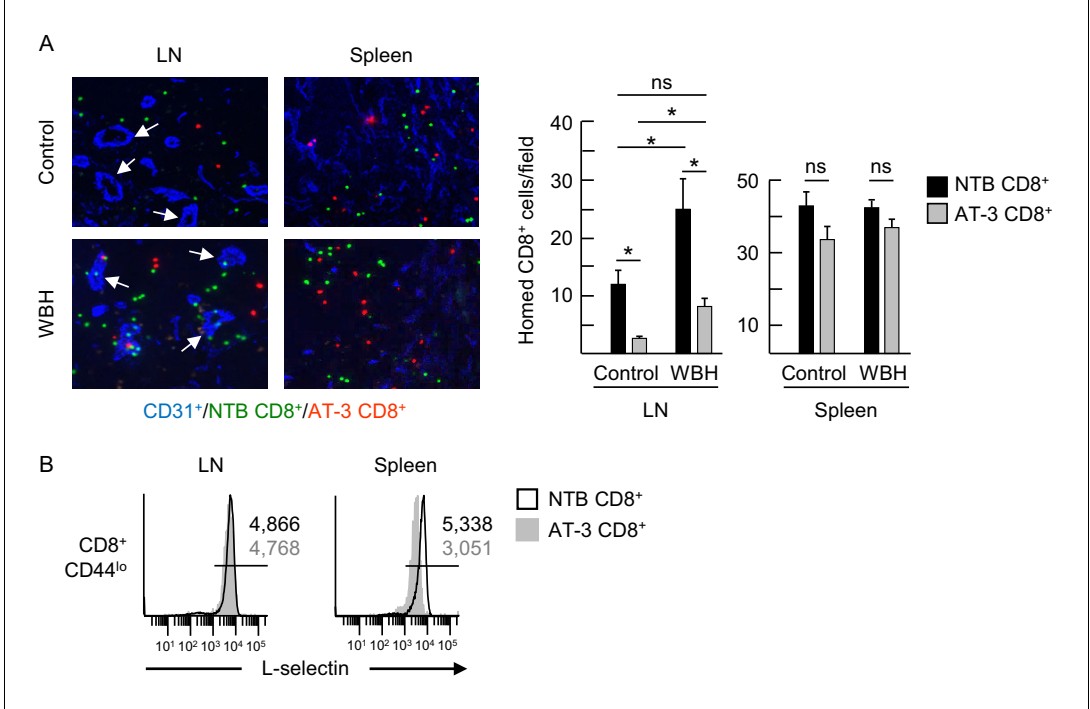

**Figure 7.** L-selectin down-modulation on CD8[+] T cells of AT-3-bearing mice inhibits trafficking across LN HEV. (**A**) Competitive homing studies used a 1:1 ratio of splenic CD8[+] T cells isolated from non-tumor bearing mice (NTB CD8[+], green) and AT-3–bearing mice (AT-3 CD8[+], red). One hour after intravenous adoptive transfer of CD8[+] T cells, the extent of homing of fluorescently-tagged transferred cells was examined in LN or spleens either of NTB controls (i.e., homeostatic trafficking) or in an inflammatory model in which the core body temperature of tumor-free recipient mice was elevated by whole body hyperthermia (WBH, 39.5 ± 0.5°C for 6 hr) prior to T cell transfer. Representative photomicrographs of fluorescently-labeled homed cells in histological LN and splenic cryosections; counterstaining with CD31 antibody identified cuboidal high endothelial venules (HEV, denoted by white arrows, *left*) in LN. Scale bar, 50 μm. Note the majority of T cells detected in images extravasated across HEV and were located in the LN parenchyma. Data (mean±s.e.m.; *right*) are from one experiment (*n* = 3 mice per group) and are representative of three independent experiments. *p<0.05; ns, not significant; data were analyzed by unpaired two-tailed Student's *t*-test. (**B**) L-selectin expression profiles of CD8[+] T cells (originating from NTB mice or AT-3–bearing mice) that were recovered 1 hr after adoptive transfer in LN or spleen of NTB recipients. Horizontal lines in histograms indicate positively stained cells; numbers are mean fluorescence intensity. NTB, non-tumor bearing; WBH, whole body hyperthermia.

The following figure supplement is available for figure 7:

**Figure supplement 1.** Adhesion molecule expression on LN HEV.

driven proliferation than OT-I cells from AT-3-bearing mice (*Figure 8B and C*). We considered that these results might be explained by T cell preconditioning (i.e., before antigen exposure) by MDSC within tumor-bearing mice. Thus, we set up parallel in vitro culture systems to model the high splenic MDSC concentrations that T cells would encounter in tumor-bearing mice (*Figure 8—figure supplement 1A*). These studies revealed that transient co-culture of MDSC with naïve OT-I CD8[+] or DO11.10 CD4[+] TcR-transgenic cells (i.e., 16 hr 'preconditioning' phase), followed by removal of MDSC, markedly suppressed T cell proliferation during subsequent challenge with cognate peptide antigens (*Figure 8—figure supplement 1B*).

We next examined the impact of L-selectin deficits on antigen-responsiveness in an in vivo model that depends on L-selectin-dependent trafficking for access to Ag. In this regard, a 1:1 ratio of L-selectin[hi] OT-I cells and L-selectin[int/lo] OT-I cells was adoptively transferred into non-tumor bearing recipients that were pre-vaccinated in the footpad with SIINFEKL-pulsed DC (*Figure 8A*). We used tumor-free recipients to interrogate the causal relationship between L-selectin loss and impaired adaptive immunity which would otherwise be difficult to assess in tumor-bearing mice because of the additional immunosuppressive mechanisms operative in LN (e.g., tolerogenic DC, Treg) (*Munn et al., 2004*; *Liu et al., 2010*). Flow cytometric analysis of transferred cells recovered from

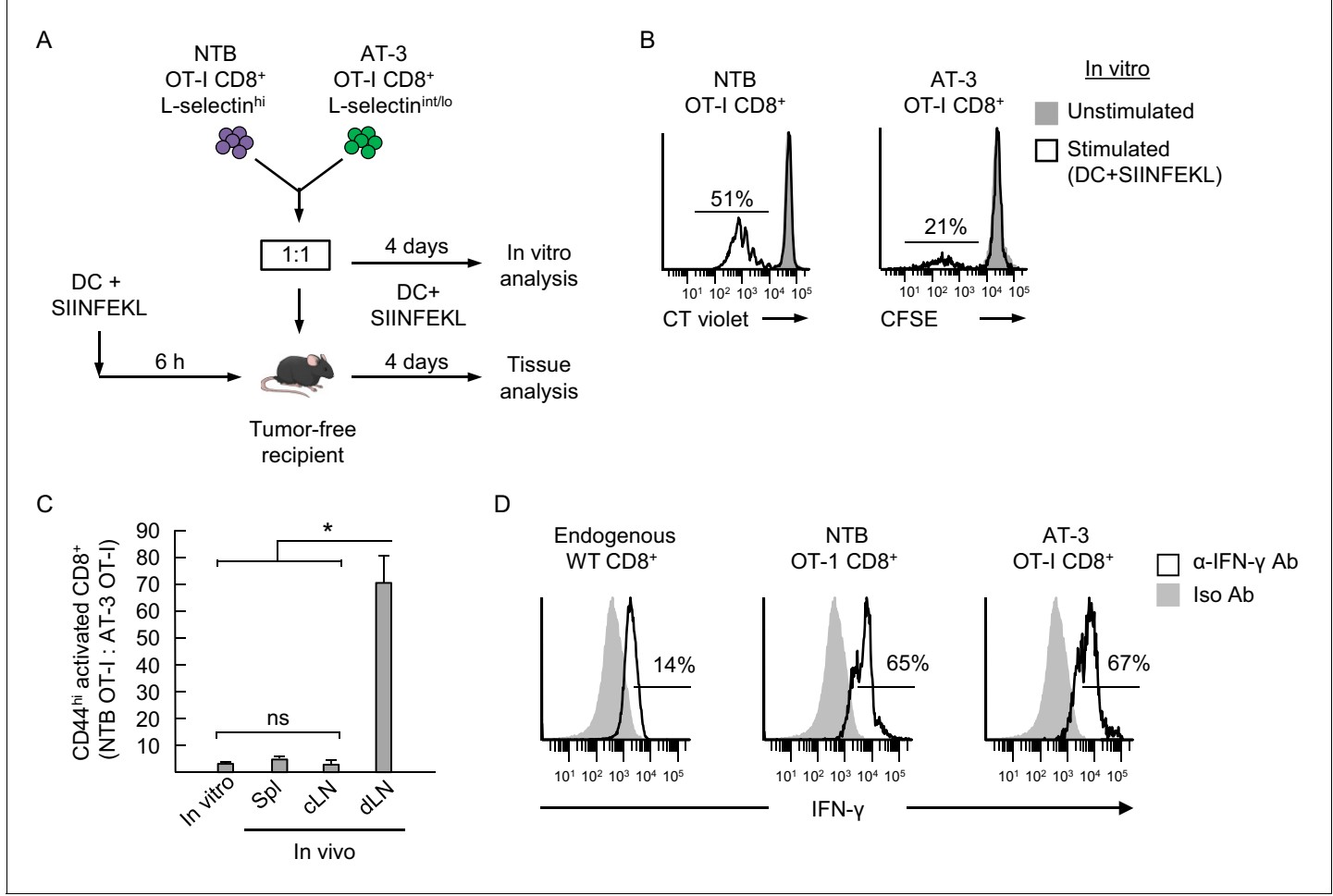

**Figure 8.** Antigen-driven activation of CD8[+] OT-I T cells is compromised by poor L-selectin-dependent trafficking in lymph nodes. (**A**) Schematic of competitive activation assays. CD8[+] T cell populations (>95% CD8[+]) were isolated from non-tumor bearing OT-I mice (NTB OT-I CD8[+] L-selectin[hi]) and from AT-3-bearing OT-1 mice (AT-3 OT-I CD8[+] L-selectin intermediate-to-low; L-selectin[int/lo]; tumor volume 3825 ± 123 mm[3] for $n$ = 4 mice). T cells (depleted of CD11b[+] MDSC) from NTB or tumor-bearing mice were then labeled ex vivo with different proliferation dyes (CellTrace Violet or CellTrace CFSE, respectively), co-mixed at a 1:1 ratio, and assessed for functional responses to cognate antigen (SIINFEKL) after four days in competitive activation assays in vitro and in vivo. (**B**) Flow cytometric analysis of proliferation of NTB OT-I CD8[+] and AT-3 OT-I CD8[+] T cells after activation by SIINFEKL-loaded dendritic cells (DC) for four days in vitro. Horizontal lines on histograms indicate percent proliferating cells. (**C**) Competitive in vivo activation assay in which NTB recipient mice were vaccinated (via footpad) with SIINFEKL-loaded DC 6 hr before adoptive transfer of a 1:1 mixture of L-selectin[hi] NTB OT-I and L-selectin[int/lo] AT-3 OT-I CD8[+] T cells. After four days, the ratios of the adoptively transferred cells were assessed in the following lymphoid compartments: spleen (Spl), contralateral popliteal lymph node (cLN), and draining popliteal lymph node (dLN). Data (mean±s.e.m.) are from one experiment ($n$ = 4 mice per group) and are representative of 2 independent experiments. *$p<0.05$; ns, not significant; data were analyzed by unpaired two-tailed Student's $t$-test. (**D**) IFN-γ expression profiles for endogenous CD8[+] T cells and adoptively transferred NTB OT-I CD8[+] and AT-3 OT-I CD8[+] T cells recovered in dLN of DC-vaccinated mice. Horizontal lines on histograms indicate positively stained cells; data are representative of 2 mice per group. NTB, non-tumor bearing; DC, dendritic cell; Iso, Isotype; Ab, antibody.

The following figure supplement is available for figure 8:

**Figure supplement 1.** Preconditioning of antigen-inexperienced TcR-transgenic CD8[+] and CD4[+] T cells with MDSC in vitro suppresses responsiveness to subsequent antigen challenge.

the draining popliteal LN (dLN) four days after adoptive transfer revealed that activated OT-I T cells from tumor-free donors outnumbered OT-I cells from AT-3–bearing donors by ~70:1 (*Figure 8C*). These OT-I cells were largely differentiated effectors based on interferon-γ production (*Figure 8D*). This biased response by L-selectin[hi] OT-I T cells was not observed outside the primary site of antigen exposure including contralateral LN (cLN) or spleen (*Figure 8C*). Taken together, the profound

discrepancy between antigen responsiveness of L-selectin$^{hi}$ versus L-selectin$^{int/lo}$ OT-I T cells in vitro and in vivo (2:1 versus 70:1 ratio, respectively; *Figure 8B and C*) supports a model in which MDSC operate at distal sites (i.e., blood and spleen compartments) to subvert adaptive immunity by restricting L-selectin-directed access of naïve CD8$^+$ T cells to cognate antigens within the LN micro-environment (*Figure 9*).

## Discussion

Regional LN are major lines of defense against cancer, serving as hubs for the generation of acute antitumor adaptive immunity and durable memory. A rate-limiting step for immune surveillance involves trafficking at HEV which ensures that DC present cognate antigens to a sufficiently diverse repertoire of naïve T cells in order to drive expansion of CD4$^+$ and CD8$^+$ effector T cell pools and B

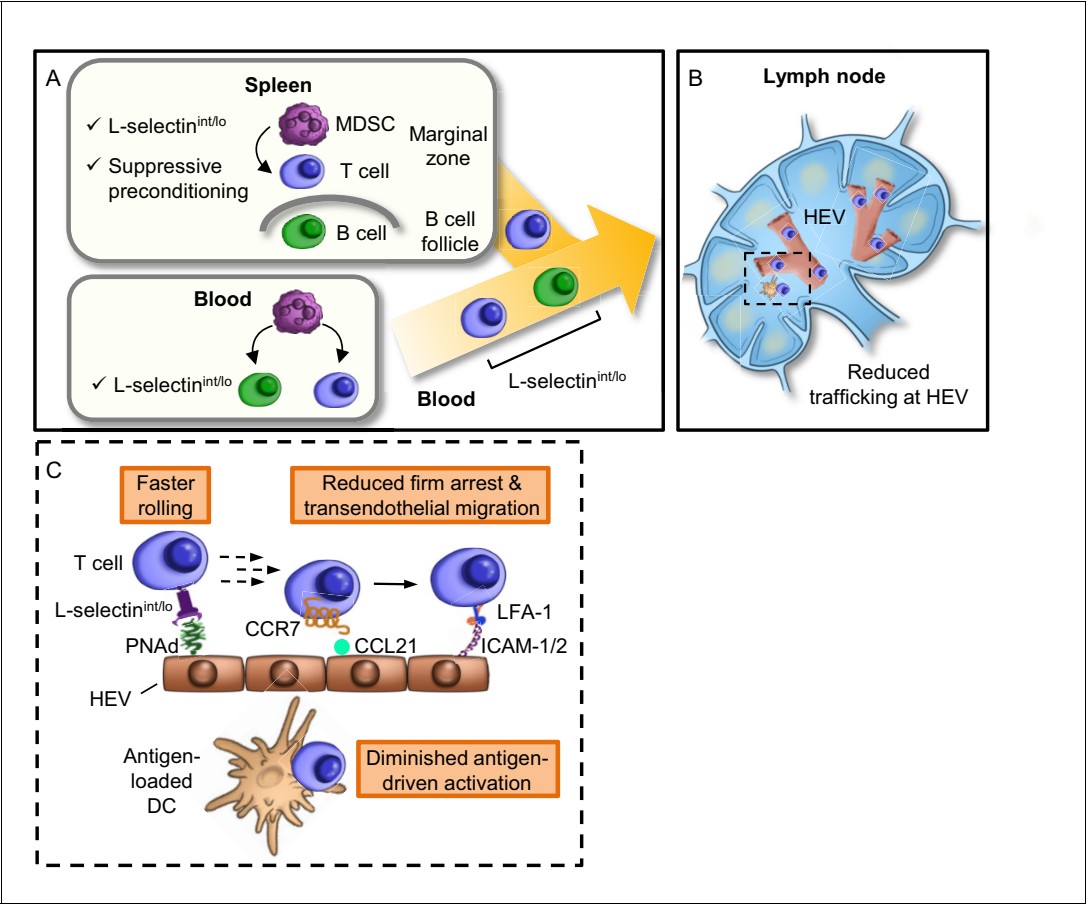

**Figure 9.** Model for MDSC actions at remote sites that compromise adaptive immunity in the LN compartment. (**A**) Restricted localization of MDSC in the splenic marginal zone leads to preferential, downregulation of L-selectin (i.e., intermediate-to-low phenotype, L-selectin$^{int/lo}$), on naïve CD4$^+$ and CD8$^+$ T cells, but not on B220$^+$ B cells. MDSC in the splenic marginal zone also precondition CD4$^+$ and CD8$^+$ T cells which leads to suppressed responsiveness to antigen outside the splenic environment. L-selectin on circulating T and B cells can be independently targeted by MDSC within the blood compartment, leading to significantly elevated levels of circulating soluble L-selectin. MDSC-mediated downregulation of L-selectin is contact-dependent and occurs post-transcriptionally, but is independent of the major L-selectin sheddase, ADAM17. (**B**) Diminished L-selectin expression reduces trafficking of blood-borne lymphocytes across high endothelial venules (HEV) in the lymph node compartment. Boxed region is shown in more detail in inset. (**C**) Inset of lymph node region showing that moderate L-selectin loss (L-selectin$^{int/lo}$ phenotype) results in faster rolling of T cells on lymph node HEV which, in turn, reduces the transition to firm arrest and subsequent transendothelial migration into the underlying parenchyma. Diminished trafficking in HEV, in combination with sustained immunosuppression caused by MDSC preconditioning in the spleen, profoundly compromises the generation of effector T cells in response to cognate antigen presented by dendritic cells (DC). L-selectin$^{int/lo}$, L-selectin intermediate-to-low expression; PNAd, peripheral lymph node addressin; LFA-1, leukocyte function-adhesion molecule-1; ICAM-1/2, intercellular adhesion molecule-1 and -2.

cell antibody production (*Girard et al., 2012*; *Evans et al., 2015*). Here we provide evidence that tumor-induced MDSC act from remote sites through two independent mechanisms to suppress systemic adaptive immunity in widely dispersed LN. The major findings of the current study establish that MDSC located in anatomically discrete sub-compartments of the spleen and, unexpectedly in the blood, downregulate expression of the L-selectin LN homing receptor in murine CD4$^+$ and CD8$^+$ naïve T cell subsets and on blood-borne murine B cells. In the case of naïve CD8$^+$ T cells, even moderate L-selectin loss severely limits trafficking across HEV, causing a profound reduction in antigen-driven expansion within the LN stroma. Thus, MDSC-induced L-selectin down-modulation could significantly impair immune surveillance during the early phases of tumor escape as well as compromise immunotherapy regimens dependent on immune cell access to LN. Additionally, our results reveal that T cells can be preconditioned by MDSC in the spleen or blood, resulting in diminished responsiveness to subsequent challenge with cognate antigen. Collectively, these findings lead us to propose that LN are important sites of MDSC suppression which could not be predicted from routine profiling of immune constituents since MDSC are largely excluded from this lymphoid compartment.

Results of the current study expand our understanding of the biological role of splenic and circulating MDSC in repressing immune function in a setting of tumor progression. The spleen is already considered a major reservoir for peripheral MDSC with direct immunosuppressive activity (*Gabrilovich et al., 2012*), while the blood is not known to be an active site of MDSC function in situ. In line with prior studies (*Ugel et al., 2012*; *Zhao et al., 2012*), we found discrete compartmentalization of MDSC within the splenic marginal zone of tumor-bearing mice. This localization was linked to specific downregulation of L-selectin on T cells but not B cells, consistent with MDSC exclusion from B cell follicles. Whether this restricted spatial distribution of MDSC is a byproduct of their profuse systemic accumulation or reflects preferential trafficking and/or retention is an open question. We initially predicted that splenic MDSC act directly on naïve T cells only within the splenic stroma, which is then reflected in the blood as L-selectin$^{int/lo}$ T cells exit the spleen. However, subsequent analysis of splenectomized tumor-bearing mice revealed that the blood is a major site for MDSC-targeted L-selectin downregulation for both T and B cells. Evidence that L-selectin down-modulation is restricted to specific organs as well as sub-anatomical compartments supports a scenario in which MDSC-to-target cell contact is a prerequisite for L-selectin down-modulation, as validated by MDSC-lymphocyte co-cultures in the present study. There is precedent from other systems that biological consequences occur as a result of cell-to-cell interactions within the blood, suggesting that MDSC could also be expected to have contact-dependent intravascular activities. In this regard, leukocyte-leukocyte interactions along vessel walls contribute to recruitment at inflammatory sites (*Grailer et al., 2009*) while stable aggregates comprised of circulating tumor cells and other cell types (e.g., leukocytes, platelets, stromal cells) function as 'traveling niches' to facilitate metastasis (*Krebs et al., 2014*). Thus, the high circulating MDSC concentration in tumor-bearing mice shown in the current report and in cancer patients (e.g., human MDSC comprise up to 45% of PBMC) (*Trellakis et al., 2013*) makes serial contact with target lymphocytes plausible either in free-flowing blood, within multicellular aggregates, or in marginated pools within capillaries, sinusoids or intravascular niches in the lung where naïve lymphocytes are transiently retained (*Galkina et al., 2005*). Regardless of whether the loss occurs in spleen or blood, L-selectin status is a critical determinant of homing potential since lymphocytes emigrating from each of these sites have nearly immediate access to LN HEV (*Chen et al., 2006*).

The extent of the homing defect in HEV is remarkable considering the modest change in overall L-selectin surface density observed for T cells in the AT-3 tumor model. During leukocyte rolling it is estimated that ~5 microvillous tips are in simultaneous contact with endothelial surfaces, with L-selectin concentration exceeding 90,000 molecules per µm$^2$ at a single microvillous tip (*Shao and Hochmuth, 1999*). Our results suggest that relatively subtle decreases in L-selectin density within these multivalent focal patches increase rolling speed, thus destabilizing adhesion within HEV. These data are reminiscent of similar observations for diminished rolling quality due to intermediate L-selectin downregulation in genetically engineered L-selectin$^{+/-}$ T cells (*Tang et al., 1998*; *Gauguet et al., 2004*). Our intravital data demonstrate that L-selectin modulation by MDSC is of primary importance during adhesion on the lumenal surface of HEV prior to extravasation. These findings do not, however, exclude the possibility that MDSC-mediated L-selectin loss also has downstream consequences in light of in vitro studies implicating extended roles for L-selectin in

controlling the polarity of monocytes necessary for transendothelial migration (*Rzeniewicz et al., 2015*) as well as antigen-driven acquisition of cytolytic function by CD8[+] T cells (*Yang et al., 2011*).

The current results establish an unprecedented correlation between MDSC expansion, L-selectin loss by T and B lymphocytes, and a >2 fold increase in circulating sL-selectin in tumor-bearing mice. Emerging clinical studies have reported that serum sL-selectin levels are also increased in patients with solid tumors (i.e., bladder and thyroid cancer), although links to MDSC were not explored (*Choudhary et al., 2015*; *Kobawala et al., 2016*). Lymphocytes are the primary source of the circulating sL-selectin pool under normal conditions, as demonstrated in studies showing that serum sL-selectin concentrations are reduced ~70% in lymphocyte-deficient $Rag2^{-/-}$ mice (*Tu et al., 2002*). Thus, increased serum sL-selectin in tumor-bearing mice likely reflects MDSC-induced cleavage from lymphocyte surface membranes although MDSC could be an additional source of sL-selectin. Accordingly, two independent but complementary mechanisms are proposed to reduce LN trafficking in cancer: (1) MDSC-directed downregulation of lymphocyte L-selectin which we showed negatively impacts homing in HEV, and (2) elevated circulating sL-selectin which, at the concentrations detected in tumor-bearing mice (1 µg/mL in tumor-bearing mice compared to ~0.4 µg/mL in tumor-free controls), reportedly functions as a competitive antagonist of L-selectin-directed lymphocyte homing in vivo. In this regard, elevation of circulating sL-selectin levels in L-selectin$^{-/-}$ mice to ~0.9 and 1.5 µg/mL by serum infusion reduces lymphocyte trafficking to LN HEV by ~40% and 65%, respectively, while concentrations of ~0.6 µg/mL are not inhibitory (*Tu et al., 2002*).

Surprisingly, MDSC-induced L-selectin downregulation does not depend on the main L-selectin sheddase, ADAM17, acting either in a *cis* or *trans* orientation. In this regard, evidence that MDSC-induced L-selectin loss occurs in ADAM17-deficient lymphocytes indicates that MDSC do not activate lymphocyte-intrinsic ADAM17 to cleave L-selectin in *cis*. Nor does ADAM17 present on the surface of MDSC (*Hanson et al., 2009*; *Oh et al., 2013*; *Parker et al., 2014*) act in *trans* to target L-selectin on juxtaposed lymphocyte membranes since downregulation occurs in the presence of ADAM17-specific inhibitors and in cells expressing a mutated ADAM17 cleavage site (i.e., L(E)-selectin). The mechanism underlying MDSC-induced L-selectin modulation closely parallels in vitro observations for an ADAM17 and ADAM10-independent ecto-protease during constitutive or inducible L-selectin shedding (i.e., triggered by antibody cross-linking of L-selectin and TcR) that is unaffected by mutation of the ADAM17 cleavage site, ADAM17/ADAM10 inhibitors, or ADAM17-deficiency (*Stoddart et al., 1996*; *Jasuja and Mier, 2000*; *Walcheck et al., 2003*; *Li et al., 2006*). Of note, tumor-induced MDSC are enriched for multiple ecto-proteases including ADAM28, ADAM9, collagenase-3 (MMP13), stromelysin (MMP3), macrophage elastase, and leukocyte elastase (*Aliper et al., 2014*). Moreover, at least one of these proteins, recombinant stromelysin (MMP3), can function in *trans* to cleave lymphocyte L-selectin in vitro (*Preece et al., 1996*). Thus, it is tempting to speculate that one of the wide array of MDSC ecto-proteases with different structural substrate requirements from ADAM17 contribute to lymphocyte L-selectin downregulation in cancer.

Our results have broad translational implications for cancer immunotherapy including vaccines that aim to stimulate antitumor antibody production. While B cells reportedly support MDSC development during cancer progression (*Bodogai et al., 2015*), to our knowledge this is the first demonstration that B cells are targets of tumor-induced MDSC. These findings complement studies in autoimmunity showing that MDSC derived from mice with collagen-induced arthritis directly suppress B cell proliferation and antibody production in vitro (*Crook et al., 2015*). The nearly complete loss of L-selectin detected on circulating B cells in response to MDSC in our study strongly suggests that B cell homing at HEV would be effectively blocked, especially given the well-established sensitivity of B cells to perturbations in L-selectin density (*Tang et al., 1998*; *Gauguet et al., 2004*). Thus, antibody responses would be expected to be severely diminished as a result of MDSC-induced suboptimal trafficking of both naïve B cells and CD4[+] precursors of the follicular helper T cells required for T-dependent antibody production in LN. Collectively, our findings could provide an explanation for observations that patients diagnosed with premalignant advanced colonic adenomas who have elevated circulating MDSC failed to produce antibodies in response to vaccination with the MUC-1 tumor-associated antigen (*Kimura et al., 2013*).

Emerging data also highlight the importance of LN trafficking for durable responses to T cell-based ACT cancer immunotherapy. Presently, there is strong interest in developing ex vivo activation protocols that maintain expression of LN homing receptors on T cell immunotherapeutics. This is founded on recent preclinical studies demonstrating that adoptive transfer of activated

L-selectin$^+$CCR7$^+$ stem cell memory T cells (T$_{SCM}$) has greater antitumor efficacy than terminally differentiated L-selectin$^{lo}$CCR7$^{lo}$ cytolytic effectors (*Gattinoni et al., 2005*, *2011*). The superior efficacy of T$_{SCM}$ is attributed to the requirement for adoptively-transferred cells to undergo L-selectin-dependent trafficking via HEV to access intranodal growth and survival factors (*Gattinoni et al., 2005*). Of related interest are preclinical data showing that the efficacy of ACT therapy is limited by MDSC (*Alizadeh et al., 2014*). Our findings suggest that an unexpected outcome of clinical ACT immunotherapy in patients with high MDSC burdens is that a high proportion of transferred T cells will undergo two major MDSC-mediated events that impair T cell responses in LN: (1) reduced LN homing due to rapid L-selectin loss within 24 hr, and (2) immune suppression stemming from T cell preconditioning prior to direct tumor-antigen exposure. These mechanisms would necessitate the transfer of excessive numbers of ex vivo-expanded T cells to achieve therapeutic efficacy.

In summary, here we uncovered combinatorial mechanisms of systemic immune suppression whereby tumor-induced MDSC localized outside LN shape the magnitude of adaptive immune responses in the intranodal compartment. Findings that L-selectin loss is not indelibly imprinted on T cells suggest that neoadjuvant treatments that target MDSC could offer promise for immune-based cancer therapies that depend on T and B cell trafficking to lymph nodes. Additionally, these results expand general concepts about the immunobiology underlying the suppressive mechanisms of MDSC that could be operative during chronic infections and inflammatory disorders such as autoimmunity.

## Materials and methods

### Animals

Female, age-matched BALB/c and C57BL/6 mice (8–12 weeks) were from the National Cancer Institute (Fredrick, MD), Charles River (Wilmington, MA), or Taconic (Hudson, NY). Male BALB/c mice (8–12 weeks) were from Charles River. Transgenic male and female OT-I mice (B6.129S6-Rag2tm1Fwa Tg(TcraTcrb)1100Mjb or C57BL/6-Tg(TcraTcrab)1100Mjb/J) were from Taconic or Jackson Laboratory (Bar Harbor, ME); DO11.10 mice (C.Cg-Tg(DO11.11)10Dlo/J) were from Jackson Laboratory. Female severe combined immunodeficiency mice (SCID; C.B Igh-1b Icr Tac Prkdc scid) were bred in-house by the Roswell Park Department of Laboratory Animal Resources, Buffalo, NY. MMTV-PyMT/B6 transgenic female mice (MTAG; >150 days of age with multifocal mammary tumors) expressing the polyomavirus middle T antigen controlled by the MMTV-LTR promoter (*Guy et al., 1992*; *Chen et al., 2003*; *Stewart et al., 2009*; *Waight et al., 2013*) were originally a gift from S. Gendler (Mayo Clinic, Scottsdale, AZ). Female and male *Adam17$^{flox/flox}$/Vav1-Cre* conditional knockout mice (6–14 weeks) in which all leukocytes lack functional ADAM17 were generated as described (*Mishra et al., 2016*). L(E)-selectin transgenic female and male mice (eight weeks) in which substitution of the membrane-proximal L-selectin extracellular domain with E-selectin prevents L-selectin shedding by ADAM17 were described previously (*Venturi et al., 2003*), and kindly provided by T. Tedder (Duke University School of Medicine). Sex- and age-matched wildtype littermates were included as controls. All mice were maintained under specific pathogen-free conditions in accordance with approved Institutional Animal Care and Use Committees protocols at participating institutions.

### Tumor models

4T1, B16, and CT26 cells were cultured in complete media (RPMI 1640 supplemented with 10% FCS, 2 mM L-glutamine, 100 U ml$^{-1}$ penicillin, 50 µg ml$^{-1}$ streptomycin, and 50 µM β-mercaptoethanol [ThermoFisher, Waltham, MA]). AT-3 were cultured in complete media supplemented with 1 mM sodium pyruvate, 1% MEM non-essential amino acids, and 25 mM HEPES (ThermoFisher). Tumor cells (10$^6$) were injected *s.c.* into the left flank or into the mammary fat pad as described (*Fisher et al., 2011*; *Waight et al., 2011*, *2013*). Peripheral lymph nodes included the left/right axillary, left/right brachial and left/right inguinal lymph nodes; the tumor-draining lymph node was identified as the left inguinal lymph node using Evans blue dye. Tumor short (*l*) and long (*L*) diameters were measured, and tumor volume was calculated as $l^2 \times L/2$. Studies were completed before tumors exceeded 20 mm in any diameter or mice became moribund. MTAG mice developed up to 10

discrete mammary gland tumors and no single tumor mass was allowed to exceed 20 mm in any diameter; total tumor burden was between 1000–12,000 mm$^3$ in individual mice.

## Flow cytometry

Analysis of single cell suspensions was performed as described (*Chen et al., 2006*; *Hanson et al., 2009*; *Fisher et al., 2011*; *Waight et al., 2011*, *2013*) using multiparameter flow cytometry with monoclonal antibodies listed in *Supplementary file 1*. Annexin V (FITC) and viability reagents (Zombie UV and aqua) (Biolegend, San Diego, CA) were used to assess the percentage of early apoptotic cells (annexin V$^+$Zombie UV$^{-/dim}$) within specific tissue compartments. Flow-count fluorospheres (Beckman Coulter, Brea, CA) were used to calculate absolute numbers of viable cells. A LSR Fortessa or LSR2 flow cytometer (BD Biosciences, San Jose, CA) was used for flow cytometric analysis; compensation and analysis were performed using Winlist 7.1 or 8.0 (Verity Software House, Inc., Topsham, ME).

## Immunofluorescence histology

Spleens were frozen in optimum cutting temperature compound (Sakura Finetek, Torrance, CA) and 9 µm cryosections were fixed at room temperature in 4% formaldehyde for 10 min. Sections were blocked 1 hr with 5% FBS and 5% rat gamma globulin (Jackson ImmunoResearch, West Grove, PA) and stained overnight at 4°C with fluorochrome-labled anti-Gr-1, anti-B220, and anti-CD3 antibodies (see *Supplementary file 1*). Quantification of relative trafficking molecule expression on LN HEV was performed as described (*Chen et al., 2006*; *Fisher et al., 2011*) in mice injected via tail vein with anti-CCL21 or anti-ICAM-1 antibodies (*Supplementary file 1*) 20 min before tissue harvest. Tissue cryosections were counter-stained with primary monoclonal anti-PNAd or anti-CD31 antibodies and fluorochrome-labeled secondary antibodies (i.e., goat anti-rat IgG, goat anti-Armenian hamster IgG, and goat anti-rat IgM; see *Supplementary file 1*). Digital images of ≥10 randomly selected fields per section (unit area of each field, 0.34 mm$^2$) were captured by observers blinded to specimen identity using an Olympus BX50 upright fluorescence microscope (Olympus Optical, Miami, FL) equipped with a SPOT RT camera (Diagnostic Instruments, Sterling Heights, MI); identical exposure times and image settings were used within each experiment. ImageJ software (http://rsb.info.nih.gov/ij) was used to quantify the relative fluorescence intensity for HEV staining as described (*Abramoff et al., 2004*; *Chen et al., 2006*; *Fisher et al., 2011*).

## T cell suppression assays

To evaluate T cell suppression during continuous exposure to MDSC, splenic CD11b$^+$Gr-1$^+$ cells were purified from non-tumor bearing (NTB) and 4T1-bearing BALB/c mice (tumor volume >2000 mm$^3$) using anti-CD11b$^+$ magnetic beads (Miltenyi Biotec, San Diego, CA; *Supplementary file 1*) as described (*Waight et al., 2011*). Isolated MDSC populations were ≥95% CD11b$^+$Gr-1$^+$ while >80% of the enriched NTB CD11b$^+$ cells coexpressed Gr-1$^+$. CD11b$^+$Gr-1$^+$ cells were combined at the indicated ratios with CFSE-labeled (ThermoFisher) splenocytes from tumor-free BALB/c mice and cultured for 72 hr with anti-CD3/CD28 antibody-conjugated beads (1 µl per 100 µl culture, ThermoFisher) and IL-2 (30 U/mL; Peprotech, Rocky Hill, NJ). T cell proliferation was measured based on CFSE-dilution determined by flow cytometry: % suppression=[1 - (proliferation with MDSC)/(mean proliferation without MDSC)] x 100.

To assess the suppressive effects of MDSC on T cells prior to exposure to antigen, splenocytes from OT-I and DO11.10 transgenic mice were 'preconditioned' by incubation overnight with MDSC from the blood of 4T1-tumor-bearing BALB/c mice (>92% CD11b$^+$Gr1$^+$ cells) (2×10$^6$ OT1 splenocytes ± 2-6×10$^6$ MDSC/4 ml; MDSC:splenocyte ratios of 1:1 or 3:1). Alternatively transgenic T cells were cultured overnight in serum-free medium (HL-1; Lonza Scientific, Walkersville, MD) without MDSC. Six replicates per sample were then pooled and resuspended in 250 µl PBS-2% FCS, and incubated at room temperature for 10 min with biotinylated antibody to Gr-1 (*Supplementary file 1*). MDSC were then depleted using Rapidsphere streptavidin magnetic beads according to the manufacturer's protocol (31.25 µl beads; StemCell Inc., Newark, CA). Depleted populations ranged from 0.4–4% CD11b$^+$Gr-1$^+$ cells. Resulting splenocytes were then cultured for three days with or without cognate peptide (5 µg of OVA$_{257-264}$ [SIINFEKL] or OVA$_{323-339}$ [ISQAVHAAHAEINEAGR] for OT-I and DO11.10, respectively; University of Maryland, Baltimore Biopolymer Core facility,

Baltimore, MD), pulsed with $^3$H-thymidine (1 μCi/50 μl/well; MP Biologicals, Santa Ana, CA) and harvested 16 hr later as described (*Hanson et al., 2009*; *Parker et al., 2014*).

## Splenectomy

BALB/c mice were anesthetized with inhalational isoflurane gas (4% for induction, 1.5% maintenance). Splenectomy and sham surgeries were performed via a left subcostal laparotomy incision as described (*Cortez-Retamozo et al., 2012*) either 10 days before or 14 days post-4T1 tumor implantation. The spleen was mobilized outside the abdomen and splenic vessels were cauterized. Buprenorphine (0.05 mg/kg) was administered *i.p.* for post-operative pain.

## Isolation of CD8$^+$ T cells for adoptive transfer

Mouse splenocytes were first subjected to CD11b$^+$ depletion using anti-CD11b$^+$ magnetic beads (Miltenyi Biotec; *Supplementary file 1*) unless otherwise indicated. To obtain an enriched CD8$^+$ T cell population (routinely > 90%), the CD11b$^{neg}$ fraction of splenocytes was then subjected to negative selection using a CD8$^+$ magnetic bead separation kit (Miltenyi Biotec; *Supplementary file 1*). Isolated CD8$^+$ T cells were labeled with CFSE, CellTrace Violet, CellTracker Orange, or calcein (ThermoFisher) prior to intravenous adoptive transfer (1.5–4 × 10$^7$ cells) into recipient mice.

## In vivo depletion of MDSC

Mice were injected *i.p.* with anti-Gr-1 antibody or isotype control antibody (*Supplementary file 1*) at three day intervals starting at day three post-tumor implantation as similarly described (*Waight et al., 2011*). MDSC depletion was assessed by flow cytometric analysis.

## In vivo and in vitro L-selectin recovery

Splenocytes from non-tumor bearing and 4T1-bearing mice (tumor volume >1000 mm$^3$) were depleted of CD11b$^+$ cells using anti-CD11b$^+$ magnetic beads (<5% residual CD11b$^+$Gr-1$^+$; Miltenyi Biotec). CD11b$^+$-depleted splenic cell populations were then labeled either with CellTrace CFSE or CellTrace Violet (ThermoFisher), respectively and co-mixed at a 1:1 ratio. For in vitro studies, the co-mixture was cultured at a concentration of 10$^6$ cells ml$^{-1}$ in complete media supplemented with 1 mM sodium pyruvate, MEM non-essential amino acids, and 25 mM HEPES. For in vivo recovery studies, 2–3 × 10$^7$ cells of the co-mixture were adoptively transferred via the tail vein into non-tumor bearing recipients. L-selectin expression on CD4$^+$ and CD8$^+$ T cells was determined by flow cytometric analysis prior to transfer (input) or four days after culture or adoptive transfer into mice.

## Adoptive transfer of human peripheral blood lymphocytes

Human peripheral blood-derived mononuclear cells from anonymous de-identified normal donors were collected from TRIMA leukoreduction filters (Trima Accel Collection System; CaridianBCT, Inc., Lakewood, CO) obtained from the Roswell Park Cancer Institute Pheresis Facility as per an approved Roswell Park Cancer Institute Review Board protocol. Leukocytes were isolated by Ficoll density gradient separation and monocyte-depleted lymphocytes were then prepared by cold aggregation as described (*Tario et al., 2011*). L-selectin expression on CD3$^+$CD45RA$^+$ human T cells (antibodies listed in *Supplementary file 1*) was examined by flow cytometric analysis in the blood 24 hr post-adoptive transfer of 2–4 × 10$^7$ cells via the tail vein into non-tumor bearing and 4T1-bearing SCID mice. Murine CD11b$^+$Gr-1$^+$ cell burden was determined by flow cytometric analysis and reported based on blood volume (i.e., number of CD11b$^+$Gr-1$^+$ cells per μl of blood) instead of based on the % CD45$^+$ cells since the lack of mature lymphocytes in SCID mice disproportionately skews MDSC representation within CD45$^+$ populations.

## Quantitative real-time PCR

Splenic CD4$^+$ and CD8$^+$ T cells from non-tumor bearing and 4T1-bearing mice were positively selected by anti-CD4$^+$ or anti-CD8α$^+$–conjugated microbeads, respectively, (Miltenyi Biotec; *Supplementary file 1*). Isolated T cell subsets were >90% pure. Cellular RNA was extracted using the NucleoSpin RNA kit (Machery-Nagel, Düren, Germany) according to the manufacturer's protocol. Quantification of total RNA extract was determined using the Nanodrop Lite Spectrophotometer (ThermoFisher Scientific, Waltham, MA). cDNA library was synthesized using iScript (Bio-Rad,

Hercules, CA), which was then used for PCR amplification of murine β-actin and L-selectin. qPCR was performed using SYBR Green (ThermoFisher Scientific, Waltham, MA) on a CFX Connect Real-Time System (BioRad, Hercules, CA). PCR primers for β-actin: forward primer: 5'-AGAGGGAAATCG TGCGTGAC-3'; reverse primer: 5'-CAATAGTGATGACCTGGCCGT-3' (*Body-Malapel et al., 2008*). PCR primers for L-selectin: forward primer: 5'-CCAAGTGTGCTTTCAACTGTTC-3'; reverse primer: 5'- AAAGGCTCACACTGGACCAC-3' (*Kerdiles et al., 2009*). The comparative Ct method was used to quantify L-selectin mRNA expression levels relative to endogenous β-actin.

## ELISA for sL-selectin

Mouse serum was stored at −80°C for two months. Serum levels of soluble (s)L-selectin were measured by ELISA kit (R and D systems, Minneapolis, MN) with serum matrix equalizing diluent buffer (BIO-RAD, Kidlington, Oxford, UK).

## MDSC-splenocyte co-culture assays for L-selectin modulation

Magnetic bead separation (Miltenyi Biotec; *Supplementary file 1*) was used to isolate splenic CD11b$^+$ MDSC and CD11b$^+$ control cells from 4T1-bearing and NTB mice, respectively. Target splenocytes were harvested from additional age-matched NTB mice and labeled with CellTrace Violet (ThermoFisher). In studies involving L(E)-selectin or *Adam17*$^{-/-}$ mice, splenocytes were shipped overnight at 4°C, and incubated at 37°C for a 5 hour-recovery period before initiation of experiments. Where indicated, MDSC and splenocytes were pretreated for 15 min with specific inhibitors for ADAM17 (PF-5480090, 10 μM; Pfizer, New York, NY) or ADAM17/10 (INCB7839, 20 μM; Incyte, Wilmington, DE) (*Witters et al., 2008*; *Wang et al., 2013*); inhibitors were also present throughout subsequent culture periods. MDSC (or CD11b$^+$ control cells) were combined with target splenocytes (at a 10:1 ratio; i.e., $2 \times 10^6$ myeloid cells and $2 \times 10^5$ splenocytes) in round-bottomed 96-well plates (Corning, Corning, NY) in media (complete media supplemented with 1 mM sodium pyruvate, 1% MEM non-essential amino acids, and 25 mM HEPES) with or without 20 U/mL IFN-γ (Peprotech) and 100 ng/mL LPS (Sigma-Aldrich, St. Louis, MO). Additional cultures were set up in HTS Transwell 96-well permeable support systems with 0.4 μm polycarbonate membranes (Corning) with target splenocytes at the bottom of transwells. After 24 hr, viable T and B cells were analyzed by flow cytometry; dead cells were excluded using Zombie viability dyes (Biolegend).

## PMA-induced L-selectin modulation assays

Splenocytes from wildtype NTB mice were labeled with CellTrace Violet, resuspended at a final concentration of $5 \times 10^6$ in complete media supplemented with 1 mM sodium pyruvate, 1% MEM non-essential amino acids, and 25 mM HEPES, and cultured overnight at 37°C in round-bottomed 96-well plates (Corning). Splenocytes from L(E)-selectin mice, *Adam17*$^{-/-}$ mice or littermate wildtype controls were shipped overnight at 4°C, and then cultured at 37°C for 5 hour-recovery period before initiating experiments. As designated, wildtype splenocytes were pretreated for 30 min with specific inhibitors for ADAM17 (PF-5480090, 10 μM; Pfizer) or ADAM17/10 (INCB7839, 20 μM; Incyte) (*Witters et al., 2008*; *Wang et al., 2013*) prior to addition of phorbol-12-myristate-13-acetate (PMA, 100 ng/mL; Calbiochem, San Diego, CA). After 2 hr, L-selectin expression on viable cells was assessed by flow cytometric analysis; dead cells were excluded using Zombie viability dye (Biolegend). The *cis*-acting L-selectin cleavage function of ADAM17 was verified by PMA-stimulation (100 ng/mL, 2 hr) of co-cultures of wildtype splenocytes and fluorescently-labeled *Adam17*$^{-/-}$ splenocytes (both from NTB mice; cultured at 10:1 ratio).

## Intravital microscopy

Intravital microscopy of inguinal lymph nodes of non-tumor bearing mice was performed as described (*Gauguet et al., 2004*; *Chen et al., 2006*). Briefly, mice were anesthetized (1 mg ml$^{-1}$ xylazine and 10 mg ml$^{-1}$ ketamine; 10 ml kg$^{-1}$, *i.p.*) and a catheter was inserted into the right femoral artery for the delivery of adoptively-transferred calcein-labeled (ThermoFisher) CD8$^+$ T cells purified from spleens of non-tumor bearing mice or AT-3-bearing C57BL/6 mice. An abdominal skin flap was made to expose the left inguinal lymph node. CD8$^+$ T cell interactions within postcapillary vessel walls were visualized with a customized Olympus BX51WI epi-illumination intravital microscopy system (Spectra Services, Ontario, NY). Rolling fraction, sticking fraction, and rolling velocity were

determined as described using off-line measurements (*Gauguet et al., 2004*; *Chen et al., 2006*). The rolling fraction was defined as the percentage of total cells that transiently interacted with vessels during the observation period. The sticking fraction was defined as the percentage of rolling cells that adhered to vessel walls for $\geq$30 s. Rolling velocities in order V venules were determined using ImageJ software (http://rsb.info.nih.gov/ij) (*Abramoff et al., 2004*) to measure the distances traveled by rolling cells over time.

## Chemotaxis transwell assay

CD8$^+$ T cells were negatively selected from the spleens of non-tumor bearing and AT-3-bearing mice by magnetic bead separation (Miltenyi Biotec; *Supplementary file 1*). Purity of isolated populations was >90%. Chemotaxis was assayed in 24 well plates with 5 µm pore polycarbonate membranes (Corning) as described (*Mikucki et al., 2015*). Media alone (complete media supplemented with 1 mM sodium pyruvate, 1% MEM non-essential amino acids, and 25 mM HEPES) or media containing 70 nM of recombinant murine CCL21 (Peprotech) was placed in the bottom chamber. CD8$^+$ T cell migration was quantified after 3 hr using a hemocytometer. Spontaneous migration was subtracted from all conditions, and data are reported as percentage of input cells for triplicates.

## Homotypic aggregation assay

Splenic CD8$^+$ T cells isolated by negative selection (Miltenyi Biotec; *Supplementary file 1*) were tested for LFA-1 function by assessing PMA-induced homotypic aggregation as previously described (*Isobe and Nakashima, 1991*). Briefly, T cells at a concentration of $5 \times 10^6$ cells/mL in complete media supplemented with 1 mM sodium pyruvate, 1% MEM non-essential amino acids, and 25 mM HEPES (in flat-bottomed 96-well plates, Corning) were pretreated for 15 min with anti-CD11a blocking antibody specific for the $\alpha_L$ subunit of LFA-1 (10 µg/mL; BD Biosciences). Cells were then treated with or without PMA (50 ng/mL; Calbiochem) for 18 hr. Cell aggregation was photographed using an inverted microscope. Cells were gently resuspended and counted by hemocytometer. Percentage aggregation = 100 x [1 – (number of free cells)/(number of input cells).

## Whole body hyperthermia

Non-tumor bearing C57BL/6 mice were treated with fever-range whole body hyperthermia (WBH; core temperature elevated to 39.5 ± 0.5°C for 6 hr), and allowed to return to baseline temperatures over 20 min before adoptive transfer of CD8$^+$ T cells as described (*Chen et al., 2006*; *Fisher et al., 2011*).

## Competitive short-term T cell homing assays

CD8$^+$ T cells from non-tumor bearing mice and AT-3 tumor-bearing C57BL/6 mice were labeled with CFSE or CellTracker Orange (ThermoFisher), respectively. Labeled T cells were co-mixed at a 1:1 ratio and adoptively transferred via the tail vein into control and WBH-treated tumor-free mice. Peripheral lymph nodes and spleens were harvested 1 hr after adoptive transfer and frozen in OCT compound (Sakura Finetek) for further analysis via immunofluorescence histology; quantification of cells that homed to LN was performed as described (*Chen et al., 2006*; *Fisher et al., 2011*). Briefly, tissue cryosections (9 µm) were counterstained with anti-CD31 antibody (*Supplementary file 1*) to demark the position of vessels; HEV were identified based on CD31$^+$ expression and cuboidal phenotype. Digital images were captured by observers blinded to specimen identity using an Olympus BX50 upright fluorescence microscope (Olympus Optical, Miami, FL) equipped with a SPOT RT camera (Diagnostic Instruments); all images were captured with the same settings and exposure time. The number of CFSE and CellTracker Orange-labeled cells were quantified in >10 fields (unit area per field, 0.34 mm$^2$).

## In vitro and in vivo competitive T cell activation assays with DC

Bone marrow-derived dendritic cells (DC) were generated as described (*Mikucki et al., 2015*) by culturing bone marrow cells (from C57BL/6 mice) for eight days in complete media supplemented with 1 mM sodium pyruvate, MEM non-essential amino acids, 25 mM HEPES, and murine granulocyte–macrophage colony-stimulating factor (~20 ng ml$^{-1}$; provided by Dr. Kelvin Lee, Roswell Park Cancer Institute). DC were matured by addition of lipopolysaccharide (0.5 µg ml$^{-1}$; Sigma-Aldrich, St. Louis,

MO) overnight and then pulsed with 5 µM OVA$_{257-264}$ peptide (SIINFEKL; InvivoGen, San Diego, CA). For in vitro activation studies, SIINFEKL-loaded DC were cultured for four days with a 1:1 co-mixture of L-selectin$^{hi}$ (from non-tumor bearing mice) and L-selectin$^{lo}$ OT-I T cells (from AT-3–bearing mice). For in vivo studies, vaccination was performed by injecting $3 \times 10^6$ peptide-loaded DC in the left hind footpad of tumor-free C57BL/6 mice. Six hours post-vaccination, mice were adoptively transferred with a 1:1 mixture of CellTrace violet-labeled (ThermoFisher) L-selectin$^{hi}$ and CFSE-labeled (ThermoFisher) CD8$^+$ T cells isolated from tumor-free and AT-3-bearing OT-I mice, respectively. Organs (draining popliteal LN, confirmed by Evans blue dye; contralateral popliteal LN; and spleen) were harvested four days-post-adoptive transfer. Flow cytometry was used to analyze OT-I T cell proliferation (based on dye dilution), activation (CD44$^{hi}$ phenotype), and differentiation (intracellular IFN-γ) as described (*Fisher et al., 2011*; *Mikucki et al., 2015*).

## Statistics

All data are shown as mean±s.e.m. and group differences were calculated by 2-tailed unpaired Student's *t* test unless otherwise indicated. Distributions of T cell rolling velocities in velocity histograms were evaluated by nonparametric Mann-Whitney U test as described (*Gauguet et al., 2004*). For all studies, *p-values<0.05 were considered significant.

## Acknowledgements

We thank C Netherby for technical assistance on the mammary fat pad model and M Appenheimer for critical reading of the manuscript. This work was supported by the NIH (CA79765 and AI082039 to SS Evans; T32 CA085183 to AW Ku; 5 T32 CA108456 to CA Powers; CA203348 to B Walcheck; GM021248 and CA115880 to S Ostrand-Rosenberg; CA140622 and CA172105 to SI Abrams; 1R50CA211108 to H Minderman); the University at Buffalo Mark Diamond Research Fund (to AW Ku); the Jennifer Linscott Tietgen Family Foundation (to SS Evans and JJ Skitzki); and the Breast Cancer Coalition of Rochester (to SS Evans and SI Abrams). Cytometry services were provided by the Flow and Image Cytometry Shared Resource facility at the Roswell Park Cancer Institute which is supported, in part, by the NCI Cancer Center Support Grant 5P30 CA016056.

## Additional information

### Funding

| Funder | Grant reference number | Author |
|---|---|---|
| National Institutes of Health | CA79765 | Sharon S Evans |
| National Institutes of Health | AI082039 | Sharon S Evans |
| National Institutes of Health | T32 CA085183 | Amy W Ku |
| National Institutes of Health | 5T32 CA108456 | Colin A Powers |
| National Institutes of Health | CA203348 | Bruce Walcheck |
| National Institutes of Health | GM021248 | Suzanne Ostrand-Rosenberg |
| National Institutes of Health | CA115880 | Suzanne Ostrand-Rosenberg |
| National Institutes of Health | CA140622 | Scott I Abrams |
| National Institutes of Health | CA172105 | Scott I Abrams |
| UB Mark Diamond Research Fund | | Amy W Ku |
| Jennifer Linscott Tietgen Family Foundation | | Joseph J Skitzki<br>Sharon S Evans |
| Breast Cancer Coalition of Rochester | | Scott I Abrams<br>Sharon S Evans |
| National Cancer Institute | Cancer Center Support Grant 5P30 CA016056 | Kieran O'Loughlin<br>Hans Minderman |
| National Institutes of Health | 1R50CA211108 | Hans Minderman |

The funders had no role in study design, data collection and interpretation, or the decision to submit the work for publication.

## Author contributions

AWK, JBM, KO, Conception and design, Acquisition of data, Analysis and interpretation of data, Drafting or revising the article; CAP, DTF, MNM, Acquisition of data, Analysis and interpretation of data, Drafting or revising the article; MD, APS, VKC, Acquisition of data, Analysis and interpretation of data; MK, Acquisition of data, Drafting or revising the article; HM, JJS, Conception and design, Drafting or revising the article; JM, Acquisition of data; DAS, BW, SO-R, SIA, Conception and design, Analysis and interpretation of data, Drafting or revising the article; SSE, Conception and design, Analysis and interpretation of data, Drafting or revising the article, Contributed unpublished essential data or reagents

## Author ORCIDs

Sharon S Evans, http://orcid.org/0000-0003-2958-6642

## Ethics

Animal experimentation: This study was performed in accordance with the recommendations in the NIH Guide for the Care and Use of Laboratory Animals. All of the animals were handled according to approved IACUC protocols at participating institutions (i.e., 859M and 1117M at Roswell Park Cancer Institute; SO01691417 at University of Maryland, Baltimore County; 15-16 #11 at University of Wisconsin, Milwaukee; and 1401-31272A at University of Minnesota). All surgery was performed under appropriate anesthesia and analgesia to minimize suffering and pain. The use of human PBMCs from anonymous, de-identified donors was classified as non-human subject research in accordance with federal regulations and thus not subjected to formal IRB review, but can be accessed through Roswell Park Clinical Research Services under the reference number BDR 069116.

## Additional files

### Supplementary files

• Supplementary file 1. Supporting information for antibodies used in current study. App, application; FC, flow cytometry; IF, immunofluorescence histology; Activ, T cell activation; Mag, magnetic isolation or depletion; Dep; in vivo antibody-mediated depletion.

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
