## [Decision Letter]

[Editors’ note: this article was originally rejected after discussions between the reviewers, but the authors were invited to resubmit after an appeal against the decision.]

Thank you for submitting your work entitled "Tumor-induced MDSC act via remote control to inhibit L-selectin-dependent adaptive immunity in lymph nodes" for consideration by *eLife*. Your article has been reviewed by three peer reviewers, and the evaluation has been overseen by a Reviewing Editor and Michel Nussenzweig as the Senior Editor. The reviewers have opted to remain anonymous.

Our decision has been reached after consultation between the reviewers. Based on these discussions and the individual reviews below, we regret to inform you that your work will not be considered further for publication in *eLife*.

As you can see from the included full reviews, the referees all felt that while the topic was of potential significance, the data fell short of the *eLife* standard of providing both clear mechanistic insight into the tumor effect on L-selectin or the general significance of the finding, a phenomenon already reported in the literature. The referees have provided suggestions for additional work that might address these deficiencies, but doubted that the information could be obtained in the interval granted by *eLife* for revisions.

Reviewer #1:

Ku and colleagues describe the impact of myeloid derived suppressor cells (MDSCs) on naïve lymphocytes, especially T cells, and their expression of L-selectin. Given the role of L-selectin in the homing of lymphocytes to lymphoid organs, a claim is made that MDSCs have systemic impact on dampening antigen-specific immunity in transplantable tumor models. This message is supported by data presented as indeed L-selectin is down regulated and the key figure of the manuscript (Figure 9) show dampened responses of pre-conditioned adoptively transferred T cells.

However, data are descriptive and lack critical mechanistic studies that would demonstrate how the observed phenomena are induced and maintained. MDSCs have been shown to bring down L-selectin expression on T cells in blood and lymph nodes in tumor bearing mice via ADAM17 in spleen, blood and lymphoid organs (J Immunol. 2009 Jul 15;183(2):937-44. doi: 10.4049/jimmunol.0804253. Epub 2009 Jun 24).

Figure 1–Figure 5 are somewhat confirmatory of the prior publication, Figure 6 is an add-on without much rationale or justification, Figure 7–Figure 8 confirm largely known impact of L-selectin on T cell trafficking. Understandably, the authors need to show that the knowledge in the field applies to their model but that could be condensed and more attention could be paid to the following: 1) Tumors are huge and enforced by implantation and thus it is not clear how relevant the data are to immunological scenario of a small developing tumor and events that could have taken place in draining lymph node and T cell priming. 2) It would be important to analyze the behavior of T cells and MDSCs in GEMMs or in chemically induced tumors, which might be closer to tumorigenic processes in humans. In fact sarcoma models generate T cell immunity that controls tumors in the initial phase. Thus, the impact of the work is unclear.

Reviewer #2:

This is a very well performed study to better understand the effect of MDSCs on T and B cells, known to be suppressed in the presence of MDSCs. in vivo data show the presence of MDSC in the local tumor microenvironment and the negative correlation with anti-tumor immunity. The authors use appropriate mouse tumor models to show that MDSCs are not only present at the tumor site but also systemically, especially in the spleen and the peripheral blood. This PBMC data are highly significant as it matches observations from cancer patients where increase in circulating and highly immunosuppressive MDSC have been found and reported many times.

The original observation the authors made is that in the presence of MDSCs, both circulating T cells and B cells have very low levels of L-selectin, which excludes them for trafficking into LNs, including tumor draining LNs. This appears to be the cause for low or no anti-tumor responses, which need to be started and boosted in the LNs. They provide a very good support for this by showing that in the OVA system, OVA-specific responses are compromised in the presence of MDSCs. The authors show in vitro, that co-culture of T cells with MDSCs leads to the L-selectin low phenotype, which is reversible within 24-48 hours of MDSC depletion.

While all the presented data are based on well executed experiments and support important conclusions, the manuscript leaves the reader wondering what is the mechanism by which the MDSCs do this. Elucidating the full mechanism could take a long time preventing the current data from being published in a timely fashion. If the authors have the way and reagents, mice, etc., to do only one more experiment, and do it in a reasonable amount of time, it would very much raise the importance of this manuscript. A very simple experiment that could start the discussion on the mechanism of action, could be an experiment to find out if this MDSC action requires cell to cell contact or only the soluble factors produced by MDSC. They already show that this activity is cross-species but that does not suggest (nor do they imply) a soluble factor. If this additional work cannot be performed for reasons that the authors can explain in the rebuttal, then some of their expectations could be described in the Discussion.

Reviewer #3:

In the accompanying manuscript, Ku et al. show that myeloid-derived suppressor cells (MDSC), induced by tumor growth in mouse models, results in decreased L-selectin surface levels on naïve lymphocytes. The authors provide evidence that L-selectin downregulation occurs only in T cells but not B cells, correlating with colocalization of MDSC and T cells in spleen. Yet, MDSC-triggered decrease in L-selectin surface levels are also observed in splenectomized mice, which hints to MDSC preconditioning in blood. Finally, the authors link reduced L-selectin levels to decreased homing to HEV-containing lymphoid tissue and impaired adaptive immune responses.

The observation of MDSC-triggered reversible reduction of surface L-selectin levels on naïve T cells is interesting and confirms data published by the same authors (Hanson et al., 2009; Parker et al., 2014). Yet, other data are not always clearly supporting the author's conclusions, and the focus on MDSC-triggered L-selectin decrease as major player in controlling adaptive immune responses appears not well bolstered. Also, the manuscript does not contain sufficient novel mechanistic insight on how MDSC control L-selectin levels on T cells.

The model the authors propose is that close spatial proximity between MDSCs and T cells in spleen promote decreased L-selectin levels on T but not B cells. Yet, the authors have not attempted to show close spatial organization in splenectomized mice (e.g., in liver sinusoids or lung capillaries), although both blood-borne T and B cells now show decreased L-selectin levels. This sheds doubts on the author's claim of close association as requirement for MDSC-dependent L-selectin decrease.

Along the same line, the molecular mechanism of MDSC-triggered loss of surface L-selectin remains unaddressed. It remains unclear whether this reflects transcriptional downregulation through cytokines, shedding (which should be detectable by soluble L-selectin in the supernatant) or other mechanisms. Given that the effect of MDSCs on surface L-selectin has been described before, there appears to be a lack of substantial new insights into this phenomenon.

The authors focus narrowly on L-selectin surface levels as reason for decreased T cell homing but do not address whether chemokine receptor signaling and integrin function may be impaired. This appears not unlikely since MDSC have multiple effects on T cells, as seen by their impaired activation. While the authors have checked CCR7 and LFA-1 surface levels in the supplement to Figure 7, there is some concern on the CCR7 plot. Why would there be a CCR7high population? Furthermore, the presence of CCR7 and LFA-1 receptors on the cell surface does not automatically mean they are functional. This may explain the drastically reduced sticking fraction despite fairly good rolling. Along the same line, does the MDSC-induced L-selectin loss lead to a shift in the absolute numbers of T cells in spleen versus lymph nodes or do these cells also induce apoptosis in T cells?

[Editors’ note: what now follows is the decision letter after the authors submitted for further consideration.]

Thank you for resubmitting your work entitled "Tumor-induced MDSC act via remote control to inhibit L-selectin-dependent adaptive immunity in lymph nodes" for further consideration at *eLife*. Your revised article has been favorably evaluated by Michel Nussenzweig (Senior Editor), a Reviewing Editor, and one reviewer.

The manuscript has been improved but there are some remaining issues that need to be addressed before acceptance. Please include the Figure in your recent response as new supplemental data (e.g. a figure supplement) and describe the result in the main text. Please also modify the Discussion as you have suggested with respect to the second point in your response. New experiments are not required.

---

## [Author Response]

[Editors’ note: the author responses to the first round of peer review follow.]

Reviewer #1:

Ku and colleagues describe the impact of myeloid derived suppressor cells (MDSCs) on naïve lymphocytes, especially T cells, and their expression of L-selectin. Given the role of L-selectin in the homing of lymphocytes to lymphoid organs, a claim is made that MDSCs have systemic impact on dampening antigen-specific immunity in transplantable tumor models. This message is supported by data presented as indeed L-selectin is down regulated and the key figure of the manuscript (Figure 9) show dampened responses of pre-conditioned adoptively transferred T cells.

However, data are descriptive and lack critical mechanistic studies that would demonstrate how the observed phenomena are induced and maintained. MDSCs have been shown to bring down L-selectin expression on T cells in blood and lymph nodes in tumor bearing mice via ADAM17 in spleen, blood and lymphoid organs (J Immunol. 2009 Jul 15;183(2):937-44. doi: 10.4049/jimmunol.0804253. Epub 2009 Jun 24).

We are grateful to the reviewer for requesting further investigation into the mechanisms underlying MDSC-induced L-selectin loss. This was a common issue raised by all three referees so we have jointly addressed all of the comments related to mechanism below. It is worthwhile noting, however, that up to this point, while a *trans*-acting MDSC-intrinsic ADAM17 mechanism has been strongly implicated in reducing L-selectin on juxtaposed T cells based on ADAM17 surface expression by this myeloid population (Hanson et al., J Immunol 2009; Oh et al., Breast Cancer Res 15:R79, 2013; Parker et al., Cancer Res 2014), this hypothesis has never been formally investigated. Conversely, there has been substantial evidence that ADAM17, which is the major known L-selectin sheddase, acts in *cis* to cleave L-selectin and other substrates (i.e., ectoenzyme and substrate located on the same cell membrane). We have clarified these points in our revised report (subsection “MDSC cause L-selectin loss through a contact-mediated mechanism independent of ADAM17”, second paragraph; Discussion, fifth paragraph). These observations prompted our systematic investigation of the mechanisms underlying MDSC-induced L-selectin loss including the role of ADAM17 in this process both in vitro and in vivo. Our new findings (described in detail below) are consistent with a sheddase-dependent mechanism but definitively rule out ADAM17 in either a *trans* or *cis* orientation in reducing lymphocyte L-selectin density. These new mechanistic results are described in the last two paragraphs of the aforementioned subsection and in the fourth and fifth paragraphs of the Discussion and we have dedicated one-and-a-half *new* main figures to these important findings (revised Figure 4, Figure 5; Figure 5—figure supplement 1). Collectively, we believe these data provide key insight into the mechanisms by which MDSC impede immune surveillance in cancer.

The principal findings of our new mechanistic studies are summarized below:

A) We established that MDSC-induced L-selectin downregulation does not involve transcriptional repression, as evidenced by the failure to alter L-selectin mRNA that was determined by quantitative RT-PCR in naïve T cell populations of tumor-bearing mice with a high MDSC burden (Figure 5; subsection “MDSC cause L-selectin loss through a contact-mediated mechanism independent of ADAM17”, second paragraph).

B) Instead, MDSC-induced L-selectin loss in tumor-bearing mice is accompanied by a profound (2.5-fold) and significant increase in the concentration of circulating L-selectin (Figure 5). These findings are in line with a sheddase mechanism operative in vivo that downregulates L-selectin on the surface of target naïve T and B lymphocytes. As discussed in the fourth paragraph of the Discussion, these data are also suggestive of yet another mechanism that leads to immune subversion since soluble L-selectin at the concentrations detected in the serum of tumor-bearing mice is known to act as a competitive antagonist of L-selectin function in vitro and in vivo.

C) We showed that a purified population (≥95% CD11b^+^ Gr-1^+^) of tumor-induced MDSC derived from 4T1-bearing mice could act directly on lymphocytes in vitro to reduce their membrane L-selectin expression (Figure 4; subsection “MDSC cause L-selectin loss through a contact-mediated mechanism independent of ADAM17”, first paragraph). This mechanism was further shown to be contact-dependent in cell-impermeable transwell assays.

D) As mentioned in our overview above, reports that MDSC express surface ADAM17 (Hanson et al., J Immunol 2009; Oh et al., Breast Cancer Res 15:R79, 2013; Parker et al., Cancer Res 2014) have implicated an MDSC-intrinsic ADAM17 *trans*-acting mechanism for cleavage of L-selectin from the surface of juxtaposed lymphocytes, as proposed in recent reviews (e.g., Gabrilovich et al., Nature Rev Immunol, 2012; Botta et al., Front. Oncol., 2014; Kumar et al., Trends in Immunology 2016). Notably, ADAM17 is the major known L-selectin ectosheddase although it reportedly targets substrates including L-selectin in *cis* as discussed in the second paragraph of the subsection “MDSC cause L-selectin loss through a contact-mediated mechanism independent of ADAM17”. These observations prompted our systematic investigation of the role of ADAM17 in MDSC-induced L-selectin modulation. This investigation involved new collaborations with Drs. Bruce Walcheck and Douglas A. Steeber who are experts in ADAM17-directed L-selectin shedding. For these studies we performed a head-to-head comparison between the well-established phorbol ester (PMA)-induced ADAM17-mediated pathway for L-selectin downregulation versus MDSC-induced L-selectin loss.

Our studies revealed a sharp demarcation in the ADAM17 requirements for PMA- versus MDSC-directed L-selectin modulation (described in subsection “MDSC cause L-selectin loss through a contact-mediated mechanism independent of ADAM17”, last paragraph; Discussion, fifth paragraph). Thus, in agreement with the obligate role of ADAM17 reported for PMA-induced L-selectin shedding, we found that PMA-induced loss in membrane-associated L-selectin was abrogated (1) by a specific pharmacological inhibitor of ADAM17 (PF-5480090) and by a dual ADAM17/10 inhibitor (INCB7839) (Figure 5); (2) in cells from transgenic L(E)-selectin mice expressing a mutated ADAM cleavage site due to substitution of the L-selectin membrane-proximal extracellular domain with the shorter E-selectin homologous domain (Figure 5); and (3) in ADAM17-deficient lymphocytes cultured either alone (Figure 5) or co-mixed with wildtype cells (Figure 5—figure supplement 1). Together these findings confirm reports of a strict requirement for *cis*-acting ADAM17 for PMA-induced L-selectin down-modulation. In contrast, MDSC-induced L-selectin downregulation in vitro was unaffected by inhibitors of ADAM17 and ADAM17/10; L(E)-selectin mutation; and lymphocyte-intrinsic ADAM17 deficiency (Figure 5). Further, while elevated constitutive L-selectin expression in mutant L(E)-selectin lymphocytes or on ADAM17^-/-^ T cells was indicative of an ADAM17 mechanism operative in vivo, this pathway was dispensable for MDSC-induced L-selectin downregulation in mutant L(E)-selectin-expressing T and B cells or in ADAM17^-/-^ cells following their adoptive transfer into MDSC^hi^ 4T1-bearing SCID mice (Figure 5). These results established that the MDSC mechanism of action surprisingly does not depend on ADAM17 acting in either a *cis* or *trans* orientation. Thus, MDSC do not stimulate lymphocyte-intrinsic ADAM17 to cleave L-selectin in *cis*. Nor do MDSC use their own ADAM17 in *trans* to target L-selectin for loss on juxtaposed lymphocyte membranes in vitro or in vivo, suggesting the involvement of a different L-selectin sheddase. The novelty of these findings and relationship to other potential mechanisms are discussed in our revised manuscript in the aforementioned paragraph.

Figure 1–Figure 5 are somewhat confirmatory of the prior publication, Figure 6 is an add-on without much rationale or justification, Figure 7–Figure 8 confirm largely known impact of L-selectin on T cell trafficking. Understandably, the authors need to show that the knowledge in the field applies to their model but that could be condensed and more attention could be paid to the following: 1) Tumors are huge and enforced by implantation and thus it is not clear how relevant the data are to immunological scenario of a small developing tumor and events that could have taken place in draining lymph node and T cell priming. 2) It would be important to analyze the behavior of T cells and MDSCs in GEMMs or in chemically induced tumors, which might be closer to tumorigenic processes in humans. In fact sarcoma models generate T cell immunity that controls tumors in the initial phase. Thus, the impact of the work is unclear.

A) We thank the reviewer for these comments and have adopted the suggestion to condense some information (detailed below) so more attention could be paid to our new mechanistic studies. Additionally, we would respectfully like to take this opportunity to clarify the novelty of our principal findings in order to explicitly address the points raised by this reviewer. The manuscript has been revised throughout to clarify these unique aspects of our findings as denoted below.

As advised by the reviewer, we extensively revised our manuscript to condense information regarding the correlation between MDSC expansion and L-selectin downregulation (Figure 1 in original submission). These data (now included in Figure 1—figure supplement 2;) demonstrate a temporal relationship between tumor growth and MDSC-induced L-selectin loss, thus supporting the relevance of our findings to small developing tumors which was an important point raised by this reviewer. Specifically, these data document that the extent of L-selectin downmodulation differs between various tumor systems and is closely correlated with MDSC burden.

We showed that MDSC located in anatomically discrete sub-compartments of the spleen (i.e., marginal zone) downregulate L-selectin expression in murine CD4^+^ and CD8^+^ naïve T cell subsets, but not on B cells (Figure 2–Figure 4 in original submission; now Figure 1). A cell-type specific function for MDSC has not previously been documented for splenic MDSC nor has the site for MDSC-induced L-selectin loss been previously investigated (Discussion, second paragraph).

Our data document that MDSC unexpectedly operate in the blood of splenectomized mice to induce L-selectin downregulation in both T and B cells (Figure 3 and Figure 4 of original submission; now Figure 2 and Figure 3). Notably the blood is not a known site of MDSC function in situ nor have B cells previously been shown to be MDSC targets in cancer (Discussion, second and sixth paragraphs).

We condensed our write-up for data involving targeting of L-selectin on human lymphocytes in xenogeneic adoptive transfer experiments (original Figure 6; current Figure 3—figure supplement 3). These results (discussed in subsection “L-selectin downregulation occurs on both T and B cells in the blood compartment”, last paragraph) validate that human cells can be targeted for L-selectin loss which has important translational implications (see Discussion, first, sixth and seventh paragraphs). These experiments also establish that the mechanisms involved are not species-restricted. Notably, results for human PBL were considered a strength of our study by reviewer #2.

We fully agree with the referee that inhibition of lymph node trafficking is highly predictable under conditions of strong L-selectin loss (i.e., 4T1 model); however the expectations are less intuitive for moderate L-selectin down-modulation as encountered early during MDSC expansion. This important point has been clarified in the revised manuscript (subsection “L-selectin loss reduces murine CD8^+^ T cell trafficking across LN HEV”, first paragraph; Discussion, third paragraph). Therefore, to address this physiologically relevant question, we used a stringent model of MDSC-induced moderate L-selectin loss (AT-3 tumor system) to identify severe limitations in naïve CD8^+^ T cell trafficking in lymph nodes via gateway high endothelial venules (HEV) (Figure 7 and Figure 8 of the original submission; now Figure 6 and Figure 7), which in turn causes a profound decrease in antigen-driven T cell expansion within the lymph node stroma (Figure 9 of the original submission;now revised Figure 8). Of note, prior reports have not investigated the impact of MDSC-induced L-selectin loss (either strong or moderate) on lymphocyte trafficking efficiency. The extent of the biological consequences of moderate L-selectin loss documented in our study was unexpected given L-selectin’s normally high density on leukocytes (>50,000 molecules per cell) which could theoretically have buffered against moderate fluctuations in expression during homing. Collectively, we believe these results, together with the new mechanistic information detailed above, provide substantial new insight in the immunobiology underlying this understudied suppressive activity of MDSC.

B) As to the second key point raised by the reviewer, we fully agree with the importance of assessing L-selectin regulation in GEMM. To this end, we have performed complementary studies assessing L-selectin expression in genetically-engineered MMTV-PyMT/B6 transgenic mice (MTAG) (Figure 3—figure supplement 2; subsection “L-selectin downregulation occurs on both T and B cells in the blood compartment”, last paragraph). These new data demonstrate that moderate L-selectin loss is evident on naïve T and B cell subsets in vivo in MTAG mice with expanded circulating MDSC populations and an overall large tumor burden. In contrast, L-selectin is unchanged in MTAG mice with smaller overall tumor burdens without MDSC expansion. We consider these data to be an important contribution to our study since they provide strong support for conclusions derived from implantable models indicating that the extent of L-selectin loss correlates with MDSC expansion, thereby suggesting that MDSC levels must exceed a threshold. Moreover, they provide a rationale for using a model with moderate L-selectin loss (i.e., the AT-3 mammary model) to interrogate the impact of L-selectin loss on T cell homing or T cell priming in lymph nodes. We have revised the manuscript to more clearly indicate that moderate L-selectin loss would be expected to be highly relevant during the early phases of tumor escape from immune surveillance as well as for immunotherapy regimens where high MDSC burdens could subvert therapy by preventing immune cell access to lymph nodes (Discussion, first paragraph).

Reviewer #2:

*This is a very well performed study to better understand the effect of MDSCs on T and B cells, known to be suppressed in the presence of MDSCs.* in vivo *data show the presence of MDSC in the local tumor microenvironment and the negative correlation with anti-tumor immunity. The authors use appropriate mouse tumor models to show that MDSCs are not only present at the tumor site but also systemically, especially in the spleen and the peripheral blood. This PBMC data are highly significant as it matches observations from cancer patients where increase in circulating and highly immunosuppressive MDSC have been found and reported many times.*

*The original observation the authors made is that in the presence of MDSCs, both circulating T cells and B cells have very low levels of L-selectin, which excludes them for trafficking into LNs, including tumor draining LNs. This appears to be the cause for low or no anti-tumor responses, which need to be started and boosted in the LNs. They provide a very good support for this by showing that in the OVA system, OVA-specific responses are compromised in the presence of MDSCs. The authors show* in vitro, that co-culture of T cells with MDSCs leads to the L-selectin low phenotype, which is reversible within 24-48 hours of MDSC depletion.

While all the presented data are based on well executed experiments and support important conclusions, the manuscript leaves the reader wondering what is the mechanism by which the MDSCs do this. Elucidating the full mechanism could take a long time preventing the current data from being published in a timely fashion. If the authors have the way and reagents, mice, etc., to do only one more experiment, and do it in a reasonable amount of time, it would very much raise the importance of this manuscript. A very simple experiment that could start the discussion on the mechanism of action, could be an experiment to find out if this MDSC action requires cell to cell contact or only the soluble factors produced by MDSC. They already show that this activity is cross-species but that does not suggest (nor do they imply) a soluble factor. If this additional work cannot be performed for reasons that the authors can explain in the rebuttal, then some of their expectations could be described in the Discussion.

We appreciate the enthusiasm by this reviewer for the overall significance and quality of our study. As stated above in our response to reviewer #1, we also fully agree with the importance of providing additional mechanistic insight into this underexplored immunosuppressive activity of MDSC. Therefore, we have performed a rigorous analysis into the underlying mechanisms which is described in our revised manuscript. In order to avoid duplication of our response since the question of mechanism was raised by all 3 reviewers, we have briefly summarized our main findings below and refer the reviewer to our more detailed response for reviewer #1/comment #1 which provides specific details about these experiments that are included in new figures (Figure 4, Figure 5). Briefly, our main findings regarding mechanism include:

A) Based on the advice provided by reviewer #2, we performed the recommended experiment to determine if MDSC action requires cell-to-cell contact. Data shown for in vitro co-cultures in transwells clearly demonstrate that direct contact between MDSC and target lymphocytes is required for the MDSC mechanism of action (Figure 4; subsection “MDSC cause L-selectin loss through a contact-mediated mechanism independent of ADAM17”, first paragraph; Discussion, second paragraph).

B) Additionally we established that MDSC-induced L-selectin down-modulation in target T or B cells: (i) does not involve transcriptional repression (Figure 5; subsection “MDSC cause L-selectin loss through a contact-mediated mechanism independent of ADAM17”, second paragraph); (ii) is accompanied by a substantial increase in circulating soluble L-selectin which is suggestive of a sheddase mechanism (Figure 5; in the aforementioned paragraph); and (iii) is not mediated by the conventional ADAM17 zinc-dependent metalloproteinase mechanism either in vitro or in vivo (Figure 5; subsection “MDSC cause L-selectin loss through a contact-mediated mechanism independent of ADAM17”, last paragraph). The latter finding was particularly surprising given that ADAM17 is the principal L-selectin sheddase identified thus far in vivo. The novel aspects of these findings and relationship to other potential mechanisms are discussed in our revised manuscript (Discussion, fourth and fifth paragraphs). Collectively, we believe these findings substantially advance current understanding of the mechanisms underlying barriers to lymphocyte access to lymph nodes that are controlled by tumor-induced MDSC.

Reviewer #3:

[…] The observation of MDSC-triggered reversible reduction of surface L-selectin levels on naïve T cells is interesting and confirms data published by the same authors (Hanson et al., 2009; Parker et al., 2014). Yet, other data are not always clearly supporting the author's conclusions, and the focus on MDSC-triggered L-selectin decrease as major player in controlling adaptive immune responses appears not well bolstered. Also, the manuscript does not contain sufficient novel mechanistic insight on how MDSC control L-selectin levels on T cells.

The model the authors propose is that close spatial proximity between MDSCs and T cells in spleen promote decreased L-selectin levels on T but not B cells. Yet, the authors have not attempted to show close spatial organization in splenectomized mice (e.g., in liver sinusoids or lung capillaries), although both blood-borne T and B cells now show decreased L-selectin levels. This sheds doubts on the author's claim of close association as requirement for MDSC-dependent L-selectin decrease.

To address this important question, we obtained new data to establish that close physical contact is a prerequisite for MDSC-dependent L-selectin loss on naïve T or B cells. Specifically, we showed that the ability of MDSC to cause L-selectin downregulation during co-culture with T cells is abrogated if effector and target cells are physically separated by cell-impermeable transwell-inserts (Figure 4; subsection “MDSC cause L-selectin loss through a contact-mediated mechanism independent of ADAM17”, first paragraph). These findings are in line with a requirement for physical interactions between MDSC and target lymphocytes in the spleen and blood to cause L-selectin downregulation. The implications of these findings are discussed in the context of a report showing that naïve T cells are retained within vascular niches in lung tissue (Galkina et al. (J Clin Invest 115:3473, 2005), suggesting that this could be a site of stable interactions between MDSC and target lymphocytes (Discussion, second paragraph). Additionally, we performed pilot feasibility studies to determine whether MDSC/T cells could be detected within lung vessels by immunohistochemical staining but were unsuccessful. In view of the extensive experimentation needed to rigorously determine if MDSC attack L-selectin within vascular niches rather than in free-flowing blood, we believe that this question is beyond the scope of the current study, particularly in light of the large body of mechanistic data now included in our report.

Along the same line, the molecular mechanism of MDSC-triggered loss of surface L-selectin remains unaddressed. It remains unclear whether this reflects transcriptional downregulation through cytokines, shedding (which should be detectable by soluble L-selectin in the supernatant) or other mechanisms. Given that the effect of MDSCs on surface L-selectin has been described before, there appears to be a lack of substantial new insights into this phenomenon.

We completely agree with this reviewer’s comments about the importance of providing substantial new insights into the mechanism of action of MDSC in triggering L-selectin loss. The question of mechanism was raised by all 3 reviewers and we have performed a rigorous series of studies to gain clarity on this issue. To avoid redundancy in our response to the 3 reviewers, below we have briefly outlined our main findings and we refer the reviewer to our response to reviewer #1/comment #1 for a more detailed explanation of our mechanistic studies. These new data are included in several new panels in our revised figures (Figure 4, Figure 5, Figure 5—figure supplement 1).

A) We performed the important experiments recommended by reviewer #3 in which we analyzed L-selectin expression by quantitative RT-PCR in CD4^+^ and CD8^+^ splenic T cells recovered from MDSC^hi^ 4T1 tumor-bearing mice (Figure 5). These data indicate that L-selectin downregulation induced by MDSC in vivo is not due to transcriptional repression in CD4^+^ and CD8^+^ T cells (subsection “MDSC cause L-selectin loss through a contact-mediated mechanism independent of ADAM17”, second paragraph).

B) We are grateful for the suggestion to perform the analysis of soluble (s)L-selectin by ELISA. We found that MDSC-induced L-selectin loss is accompanied by a profound (2.5-fold) increase in circulating sL-selectin concentrations when values were compared to non-tumor bearing mice (Figure 5; subsection “MDSC cause L-selectin loss through a contact-mediated mechanism independent of ADAM17”, second paragraph). These data are suggestive of a sheddase-dependent mechanism operative in vivo by which MDSC trigger L-selectin loss from the surface of naïve T and B cells. Additionally these findings suggest that a second mechanism impedes naïve lymphocyte trafficking to lymph nodes in tumor-bearing mice since sL-selectin concentrations in the range we identified in the serum of tumor-bearing mice (~1 µg/ml) are known to competitively antagonize L-selectin function in vitro and in vivo. The implications of these findings are discussed in the fourth paragraph of the Discussion.

C) We performed a head-to-head comparison between the conventional PMA-induced ADAM17 pathway and MDSC responses to determine if similar mechanism are used to direct L-selectin loss on target lymphocytes (Figure 5; Figure 5—figure supplement 1; subsection “MDSC cause L-selectin loss through a contact-mediated mechanism independent of ADAM17”, last paragraph). These studies used a series of complementary approaches including ADAM17/ADAM10-specific pharmacological inhibitors, lymphocytes expressing mutant L(E)-selectin that lack the ADAM-cleavage site (provided by Dr. Doug Steeber), and ADAM17-deficienct lymphocytes (provided by Dr. Bruce Walcheck). The results described in Figure 5 surprisingly revealed that MDSC-induced L-selectin downregulation in T and B cells does not depend on ADAM17 acting in either a *cis* or *trans* orientation. Thus, these findings exclude the possibility that MDSC provide extrinsic signals to activate *cis*-acting ADAM17 in lymphocytes. Nor do MDSC use their own ADAM17 in *trans* to target L-selectin for loss on juxtaposed lymphocyte membranes in vitro or in vivo. These findings point to another L-selectin sheddase and are fully discussed in the fifth paragraph of the Discussion. Collectively, we believe that these new findings substantially expand the impact of our findings by identifying an unexpected mechanism underlying MDSC regulation of L-selectin expression in naïve lymphocyte populations.

The authors focus narrowly on L-selectin surface levels as reason for decreased T cell homing but do not address whether chemokine receptor signaling and integrin function may be impaired. This appears not unlikely since MDSC have multiple effects on T cells, as seen by their impaired activation. While the authors have checked CCR7 and LFA-1 surface levels in the supplement to Figure 7, there is some concern on the CCR7 plot. Why would there be a CCR7high population? Furthermore, the presence of CCR7 and LFA-1 receptors on the cell surface does not automatically mean they are functional. This may explain the drastically reduced sticking fraction despite fairly good rolling. Along the same line, does the MDSC-induced L-selectin loss lead to a shift in the absolute numbers of T cells in spleen versus lymph nodes or do these cells also induce apoptosis in T cells?

The reviewer raised a number of important points which we have addressed as follows:

A) We agree that the CCR7 staining shown in our original submission was sub-par and we have repeated these experiments with a more optimized staining protocol. These new data show that neither CCR7 nor LFA-1 expression is altered on T cells isolated from AT-3 bearing mice, despite profound downregulation of L-selectin surface expression (subsection “L-selectin loss reduces murine CD8^+^ T cell trafficking across LN HEV”, second paragraph;Figure 6—figure supplement 1).

B) We have addressed the excellent point raised by the reviewer by expanding our analysis to include the function of other relevant trafficking molecules on purified CD8^+^ T cells isolated from tumor-bearing mice (Figure 6—figure supplement 1). The results from in vitro assays for CCL21/CCR7-driven chemotaxis in transwell assays and PMA-induced LFA-1-dependent homotypic lymphocyte aggregation assays (subsection “L-selectin loss reduces murine CD8^+^ T cell trafficking across LN HEV”, second paragraph) indicate that the function of these trafficking molecules is not compromised in cancer. Thus, L-selectin loss appears to be the primary explanation for our observations of poor trafficking in lymph node high endothelial venules.

C) We addressed the final question by performing the recommended quantification of T cell numbers in lymph nodes and spleen of non-tumor bearing control mice and 4T1-bearing mice (Figure 1). These results show that the absolute numbers of naïve CD4^+^ and CD8^+^ T cells is diminished selectively in the lymph nodes while we see a compensatory increase in the spleen of tumor-bearing mice. We did not observe any change in the apoptotic fraction of T cells at either of these tissue sites. These findings are in line with our hypothesis that MDSC-induced L-selectin loss contributes to poor overall trafficking of T cells to lymph nodes, thereby compromising immune surveillance in tumor-bearing mice (subsection “Spatiotemporal correlation between L-selectin loss and MDSC co-localization with naïve T cells in the splenic compartment”, third paragraph).

[Editors’ note: the author responses to the re-review follow.]

The manuscript has been improved but there are some remaining issues that need to be addressed before acceptance. Please include the Figure in your recent response as new supplemental data (e.g. a figure supplement) and describe the result in the main text. Please also modify the Discussion as you have suggested with respect to the second point in your response. New experiments are not required.

1) We included a new Figure (Figure 3—figure supplement 2) and described the result in the main text (subsection “L-selectin downregulation occurs on both T and B cells in the blood compartment”, third paragraph) that shows that near-complete L-selectin loss on T and B cells occurs within 2 hours after adoptive transfer into the MDSC-rich vascular compartment of 4T1 tumor-bearing mice. As discussed in the revised manuscript, these data further support our conclusion that the blood is a prominent site for MDSC-induced L-selectin loss since the lymphocytes analyzed 2 hours after transfer are those that remained in the blood for the duration of the experiment; i.e., this time period is not sufficient for blood-borne naïve lymphocytes to recirculate through tissues and return back to the blood.

2) We revised the Discussion to more fully describe various scenarios whereby physical interactions between MDSC and naïve T and B cells within the intravascular space could be expected to culminate in profound L-selectin loss (second paragraph).

3) We modified the Discussion to more explicitly describe the experiments cited (fourth paragraph) showing that soluble L-selectin at the concentration range we detected in tumor-bearing mice (~1 µg/mL) can function as a competitive antagonist of L-selectin-dependent lymphocyte trafficking to LN HEV in vivo (Tu et al., J Immunol 2002).